applied mathematics/biomechanics/ bioengineering

blood flow, Darcy, multi-scale model, perfusion, porous media, vascular system

**Author for correspondence:**
Ulin Nuha A. Qohar
e-mail: ulin.qohar@uib.no

# A nonlinear multi-scale model for blood circulation in a realistic vascular system

Ulin Nuha A. Qohar, Antonella Zanna Munthe-Kaas, Jan Martin Nordbotten and Erik Andreas Hanson

Department of Mathematics, University of Bergen, Allegaten 41, Bergen 5008, Norway

UNAQ, 0000-0003-3747-4878

In the last decade, numerical models have become an increasingly important tool in biological and medical science. Numerical simulations contribute to a deeper understanding of physiology and are a powerful tool for better diagnostics and treatment. In this paper, a nonlinear multi-scale model framework is developed for blood flow distribution in the full vascular system of an organ. We couple a quasi one-dimensional vascular graph model to represent blood flow in larger vessels and a porous media model to describe flow in smaller vessels and capillary bed. The vascular model is based on Poiseuille's Law, with pressure correction by elasticity and pressure drop estimation at vessels' junctions. The porous capillary bed is modelled as a two-compartment domain (artery and venous) using Darcy's Law. The fluid exchange between the artery and venous capillary bed compartments is defined as blood perfusion. The numerical experiments show that the proposed model for blood circulation: (i) is closely dependent on the structure and parameters of both the larger vessels and of the capillary bed, and (ii) provides a realistic blood circulation in the organ. The advantage of the proposed model is that it is complex enough to reliably capture the main underlying physiological function, yet highly flexible as it offers the possibility of incorporating various local effects. Furthermore, the numerical implementation of the model is straightforward and allows for simulations on a regular desktop computer.

## 1. Introduction

Nowadays, computational approaches have become one of the complementary tools in studying structure, function and blood regulation of the vascular systems [1–10]. The fundamental purpose of developing a numerical model is to understand how changes in the vascular structure affect transport mechanisms in organs, giving important clinical information. However, one of

the challenging modelling issues for these systems is the fact that vascular systems are made of vessels at different scales [8], ranging from large arteries with close to turbulent blood flow, to smaller arteries and arterioles with mostly laminar flow, and to capillaries, with perfusion of cellular particles between artery and vein capillaries. This complexity makes it difficult to fully understand the connection between scales and the effect of localized changes on the whole organ.

The multi-scale mathematical modelling for blood circulation offers a reasonable solution to these difficulties. By combining the well-established flow models at various vascular scale levels, one may achieve a global model that also takes into account multiple local properties. The coupling between each level is the key to creating a complete model with a suitable balance between computational cost and detail, as required for pathological characterization [6,8,11]. To this end, several earlier studies have proposed suitable modelling set-ups [1,5,7,12–16]. The authors in [5] propose a three-dimensional/one-dimensional coupling for combining large and small arteries in the cerebral vasculature. In [10], the authors consider a network representation that includes arteries, veins and capillaries. This model assumes that the whole vasculature is an interconnected tube network and results in a much larger system. However, the computational cost of this approach limits its applicability at a clinically relevant scale. To overcome the limitation, the authors in [10] replaced the capillaries and arterioles with a coarser network with similar resistivity. In a recent study [3], the authors treat vessels as a one-dimensional network model and the capillary bed as a three-dimensional continuum model, thus decreasing significantly the computational cost and allowing for full brain simulations. Other works addressing a discrete-continuum system have investigated the blood flow simulation for cerebral and liver microvasculature, both in humans and animals [7,12,17,18]. It must be mentioned, however, that zero-dimensional and one-dimensional models have a reduction in accuracy when compared to full three-dimensional models, in particular when applied to describe large vessels [19].

In the current work, we propose a multi-scale model for blood circulation that incorporates nonlinearities induced by the vascular structure. The blood flow in the vascular network (arteries and veins) is described using a vascular graph model [10] in which vessel segments are represented as long cylindrical tubes with constant radii. To compensate for the accuracy loss of this model, a model for pressure drop is included at vessel bifurcations [19,20] and we allow for a vessel radius dependence on pressure due to vessel elasticity [21]. Darcy flow is used to describe the continuum representation of the micro-circulation in the smaller vessels and the capillary bed. Further, a continuous distribution is introduced to model the unresolved structure between the vascular network terminals and the Darcy domain, with the purpose of reflecting the micro-vessel structure surrounding each terminal node. Differently from [3], we investigate the impact of nonlinear interactions between pressure and flow, as well as the role of vessel bifurcations in the flow network, and the effect of vessel occlusion in an organ. Our model was developed for full-organ simulations with parameters from experiment data on frogs [22–25] accompanied by estimations [21,26–28] to obtain results purely from governing equations and the given anatomy.

# 2. Material and methods

In this section, we present our modelling framework. We construct a system consisting of an arterial and venous vascular network, a two-compartment capillary bed and the couplings between the parts. A realization of the model configuration is illustrated in figure 1, where the dotted line represents the coupling and connection between compartments. Blood flows from the roots of the arterial network (bottom left) through the arterial network (red), then to the continuous domains of the capillary compartments (light red for arterial and light blue for venous compartment), and finally to the venous network structure (blue) and to the venous roots at the bottom right of the venous network. The arterial and venous terminal nodes are connected to the respective continuous capillary domains in the capillary bed (red solid and blue dashed dotted-lines).

## 2.1. Model simplifications and hypotheses

Our modelling framework is quite generic and, as such, does not take into account all micro and macro effects related to the flow in the vascular system. Yet, by expressing the system in a multi-scale framework, we manage to capture some of the underlying micro-scale details using the macro-scale parameters.

The main simplification in this model is that the capillary bed is assumed to be a homogeneous and isotropic porous media. Further, the nonlinearities induced by the capillary physiology and the blood

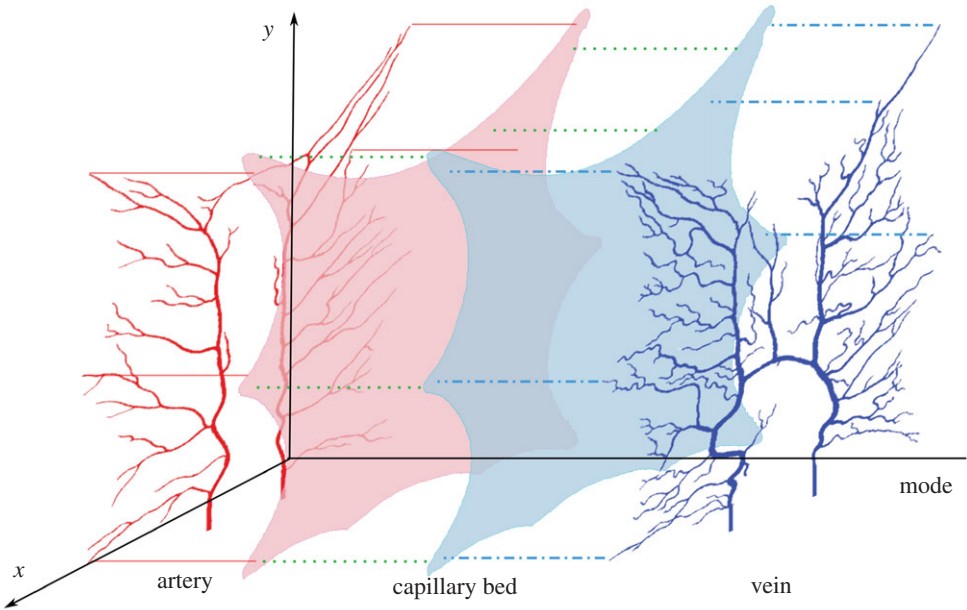

**Figure 1.** The quasi-three-dimensional numerical model for a two-dimensional spatial problem. The *xy* (vertical) axis represents the two-dimensional computational domain in space and the third axis (horizontal) is the model axis. Blood flows from the roots of the arterial network through the arterial network (red), then to the continuous domains of the capillary compartments (light red for arterial and light blue for venous compartment) and finally to the venous network structure (blue) and to the venous roots at the bottom of the venous network. The arterial and venous terminal nodes are connected to their respective continuous capillary domains in the capillary bed (red solid and blue dashed dotted lines). The green dotted lines represent the pixel-wise bridge between capillary compartments, which is modelled as the blood perfusion in §2.4.

itself are neglected [29]. These include, among others, the non-Newtonian behaviour of the blood [30], the Fahraeus–Lindqvist effect [31], plasma skimming [32] as well as viscoelasticity in the capillary bed. The effect of these limitations is partly studied in a recent simulation framework [33]. Nevertheless, the reported simulations are only performed on a micro-scale and, to our knowledge, no current simulation framework is able to incorporate all these micro-effects into a multiscale model.

With the simplifications mentioned above, the nonlinearities in this work are restricted to effects arising from and affecting the vascular network, namely pressure drop at bifurcations and vessel elasticity. Other than that, we limit our model to situations where the structure of the vascular system is in a fixed passive condition (steady-state), so that all parameters and coefficients are constant over the simulation time.

## 2.2. Graph structure model for vascular networks

A system of equations was constructed based on a network structure for both arteries and veins. Assuming laminar flow and non-slip conditions on the vessel walls, each vessel segment $n$ was modelled as a long cylindrical tube of length $L_n$ with constant radius $r_n \ll L_n$. The vessel radius was computed as an average radius value from the segmentation data. The pressure drop $\Delta P_n$ in a single vessel segment $n$ was computed using Hagen–Poiseuille's Law [10]

$$\Delta P_n^h = \frac{8\mu L_n q_n}{\pi r_n^4}, \tag{2.1}$$

where the upper index $h$ stands for hydrodynamic, $\mu$ is the blood viscosity and $q_n$ is a volumetric blood flow. At a junction node, a pressure match scheme was used to define vessel pressure drop connected to the junction node [34]. An additional pressure drop estimation was defined from [19,20]

$$\Delta P_n^b = \frac{\rho q_{dat}^2}{2\pi^2 r_{dat}^4} \left( 1 + \frac{q_n^2 r_{dat}^4}{q_{dat}^2 r_n^4} - \frac{2q_n r_{dat}^2}{q_{dat} r_n^2} \cos\left(\frac{3}{4}\theta_{(dat,n)}\right) \right), \tag{2.2}$$

where the upper index $b$ stands for bifurcation and the index dat refers to the datum vessel, i.e. the vessel from which the flow approaches the junction. Further, $\theta_{(dat,n)}$ is the angle between the datum vessel and

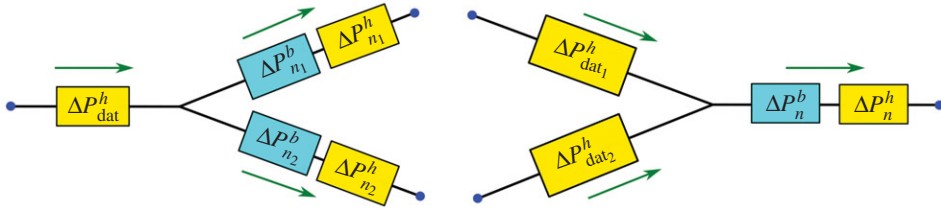

**Figure 2.** Illustration of pressure drop at junctions, see (2.3). The arrow represents the direction of blood flow.

vessel $n$. The pressure drop equation was constructed based on Bernoulli equation to match the total pressure at the inlet and outlet vessels at bifurcation. The derivation details were described in [20] for converging and diverging flow at a junction. Hence, the total pressure drop after a bifurcation node was computed as the sum of both equations (2.1) and (2.2)

$$\Delta P_n = \Delta P_n^h + \Delta P_n^b, \tag{2.3}$$

see figure 2. The other governing equation was the conservation of mass at a node

$$\sum q_{\text{in}} = \sum q_{\text{out}}, \tag{2.4}$$

with $q_{\text{in}}$ representing the blood that flows into the node and $q_{\text{out}}$ is a flow out of the node.

## 2.3. Elasticity on vessel walls

Vessel wall elasticity provides capacitance and pulse-wave dampening, allowing the arteries to maintain a relatively constant pressure despite the pulsating nature of the blood flow. On the other hand, the vessel wall elasticity also allows for a vessel radius dependence on the pressure gradient, to regulate blood supply in case of an alteration of the vascular system [21]. The change in radius due to pressure was described by

$$\Delta r = r - r_{\text{init}} = \frac{(1 - \lambda^2) r_{\text{init}}^2}{hE} (P_{\text{in}} - P_{\text{ext}}), \tag{2.5}$$

where $r_{\text{init}}$ is the initial vessel radius without pressure gradient, $h$ is the vessel wall thickness, $E$ is the Young modulus, $\lambda$ is the Poisson ratio, and $P_{\text{in}}$, $P_{\text{ext}}$ are the pressures inside and outside the vessel, respectively. The vessel thickness is assumed to be proportional to the vessel radius, with $h = 0.22r$ for the arteries and $h = 0.1r$ for the veins. This is within the approximated range reported in [24]. For the elasticity parameters, the Young modulus in the arteries $E_a$ is two times bigger than $E_v$ in the veins, $E_a = 2E_v = 1$ MPa, and $\lambda = 0.5$ [21,27]. $P_{\text{in}}$ is defined as the average pressure in the segment, that is the average pressures at both segment endpoints, and $P_{\text{ext}}$ as the capillary pressure at the midpoint outside the vessel. The elasticity equation (2.5) has the effect of introducing a nonlinearity to the system. Thus, the pressure difference between two adjacent nodes in a vessel is determined by substituting equation (2.5) into equations (2.1) and (2.2).

## 2.4. Capillary model

In the capillary model, smaller vessels and the capillary bed were discretized with a uniform grid and described by Darcy's single-phase flow equation. Darcy's Law, describing the flow of a fluid in a porous medium, states that a fluid flows from regions of higher pressure to regions of lower pressure in a linear manner. Thus

$$\mathbf{v} = -\frac{\mathbf{K}}{\mu} \nabla P, \tag{2.6}$$

where $\mathbf{v}$ is the Darcy flux (volumetric flow rate per unit area ($\text{m}^3\ \text{s}^{-1}\ \text{m}^{-2}$)), $\mathbf{K}$ is the permeability tensor of the porous medium and $\mu$ is the viscosity [35]. In addition, we assume conservation of mass (continuity equation)

$$\nabla \cdot \mathbf{v} = Q, \tag{2.7}$$

where $Q$ is the source term ($\text{s}^{-1}$). In this model, $Q$ is a source or sink, for instance describing the flow in or out a terminal node in the arterial or venous system, or describing the passage of flow from one compartment to the other. We assume blood to be an incompressible fluid, and by incorporating Darcy flow into the continuity

equation, we obtain

$$-\nabla \cdot \left(\frac{\mathbf{K}}{\mu}\nabla P\right) = Q. \tag{2.8}$$

The capillary bed is described as a two-compartment system, using one compartment for the arterial part and one for the venous part. Blood perfusion is interpreted as the interchange between the two compartments, i.e. the exchange of oxygenated blood with deoxygenated blood in the capillary bed. The driving force for perfusion is the pressure difference between the two compartments [3], giving the linear relation

$$F = \alpha(P_a - P_v), \tag{2.9}$$

where $\alpha$ is the perfusion parameter. The Darcy model for the two compartments becomes

$$\left.\begin{array}{rl} -\nabla \cdot \left(\frac{\mathbf{K}_a}{\mu}\nabla P_a\right) = Q_a - \alpha(P_a - P_v) & \text{in } \Omega_a \\ -\nabla \cdot \left(\frac{\mathbf{K}_v}{\mu}\nabla P_v\right) = Q_v + \alpha(P_a - P_v) & \text{in } \Omega_v \\ u_\beta \cdot n_\beta = 0 \quad (\text{Neumann BC}) & \text{on } \partial\Omega_\beta, \end{array}\right\} \tag{2.10}$$

and

where $\Omega_\beta$ is the Darcy capillary volume and the index $\beta = \{a, v\}$ stands for artery and vein, respectively.

## 2.5. Coupling vascular structure and capillary models

To complete the system, both the arterial and venous Darcy systems were combined with the terminal nodes of the arterial and venous networks as point sources/sinks. The analytic solution of the Darcy equation around these points is a Dirac delta function, with a singularity at the source/sink point, creating a problem at the three-dimensional–one-dimensional interface. To avoid this problem, a fluid distribution function was introduced [3]

$$f(x) = \begin{cases} C\exp\left(\frac{-1}{1-|x|^2}\right) & \text{if } |x| < 1 \\ 0 & \text{if } |x| \geq 1, \end{cases} \tag{2.11}$$

and this function was used with a finite radius $\varepsilon$, $f^\varepsilon(x) = f(x/\varepsilon)$ on the Darcy domain. The finite radius $\varepsilon$ can be set to a user-chosen constant or to a fraction of the terminal vessel's radius. In this manuscript, it is set to three pixels to represent the vessels that are not observable in the imaging data. As a result, the flow at a terminal node $Q_t$ is coupled to the capillary compartment as a number of sources/sinks $Q_\beta(x)$ according to the relation

$$Q_t^\varepsilon = \int_{\Omega_\beta} Q_\beta(x)f^\varepsilon(x - x_t)\mathrm{d}x. \tag{2.12}$$

Beside mass continuity, we have to describe pressure continuity between the one-dimensional vascular graph and the three-dimensional Darcy model. The pressure drop between the terminal node $t$ and the surrounding tissue was thus given as

$$P_t - \int_{\Omega_\beta} P_\beta(x)f^\varepsilon(x - x_l)\mathrm{d}x = \kappa Q_t, \tag{2.13}$$

where $\kappa$ represents the resistance estimation for capillary system around terminal $t$ in the Darcy model. In this work, the resistance $\kappa$ was estimated as a constant by using equation (2.1) to compute the resistance of a cylindrical tube connecting a terminal node to the Darcy domain inside the sphere of radius $\varepsilon$ centred at the terminal node. The tube has a length of $\varepsilon$ and a radius of $m^{-1/3}r_t$, where $m$ is the number of computational cells within the sphere.

Both flow and pressure couplings between the vascular graph and Darcy model close the loop from the arterial roots to the venous roots. While a well-posedness analysis of the full model with nonlinear effects is beyond the scope of this paper, a linearized model similar to the one considered in this paper was recently shown to be well posed, both in the continuous setting and when discretized by low-order finite volume methods [36].

## 2.6. Numerical implementation of the flow

### 2.6.1. The vascular network

It consists of vessel segments and nodes and is governed by (2.1), (2.4) and (2.11). For each graph segment $n$, the pressure drop (2.1) (linear part) and the pressure drop at junctions (2.2) (nonlinear part) were considered. When considering elasticity, equation (2.5) was used.

The equations at the nodes are of three different types: at the root nodes, we have control equations/ boundary conditions (given with the problem); at the internal nodes (connecting two or more segments) we have conservation of volume (2.4), and at the terminal nodes we have coupling equations with the Darcy model. In our four roots terminal (two arterial and two venous roots), there were several options to assign the boundary conditions.

Five further control equations must be considered for the system to have a unique solution. These were constructed by assigning constant pressure on the arterial and the venous root nodes (Dirichlet BC) and conservation of mass for the whole system

$$P_{\text{root},\beta} = P_{\text{BC}} \tag{2.14}$$

and

$$\sum_{n \in N_{\text{root}}} q_n = 0, \tag{2.15}$$

where root, $\beta = a, v$ is a root node on the arterial or the venous structure, $P_{\text{BC}}$ represents the constant pressure on two arterial roots (input) and two venous roots (output), and $N_{\text{root}}$ are root segments, with $q_n > 0$ for the artery and $q_n < 0$ for the vein. Finally, the whole complex is represented as a system of nonlinear equations $\mathbf{F}(\mathbf{x}) = \mathbf{b}$, where $\mathbf{F}(\mathbf{x}) = \mathbf{A}(\mathbf{x})\mathbf{x}$.

### 2.6.2. The Darcy domains (capillary bed)

The Darcy equations (2.10) in the capillary bed were solved using a two-point flux approximation (TPFA) on a uniform grid. We assume that the permeability $\mathbf{K}$ in the capillary bed is constant and uniform, and represent it by $K$ (scalar). First, we computed the fluid transmissibility between adjacent cells $i$ and $j$,

$$\tau_{ij} = 2S_{ij} \left( \frac{\Delta x_i \mu_i}{K_{\beta,i}} + \frac{\Delta x_j \mu_j}{K_{\beta,j}} \right)^{-1}, \tag{2.16}$$

where $S_{ij}$ is the face area between cells $i$ and $j$, and, as above $\beta \in \{a, v\}$. Because of the homogeneity of the system, constant parameters and uniform discretization, equation (2.16) simplifies to

$$\tau_{ij} = S \frac{K_\beta}{\mu \Delta x}. \tag{2.17}$$

Then, applying TPFA to (2.6) in single cell $i$ and adjacent cells $j \in N_i$, neighbour cells around $i$, we obtain

$$\sum_{j \in N_i} \tau_{ij}(P_i - P_j) = Q_i. \tag{2.18}$$

Applying this procedure for all cells in the domain, we obtain a system of equations $\mathbf{A}_{D-D}\mathbf{x} = \mathbf{b}$, where $\mathbf{A}_{D-D}$ is a symmetric matrix with elements

$$a_{ik} = \begin{cases} \sum_{j \in N_i} \tau_{ij} & \text{if } k = i \\ -\tau_{ik} & \text{if } k \neq i. \end{cases} \tag{2.19}$$

We note that while the TPFA method is not a consistent discretization for general grids, it is consistent and convergent on the uniform grids used in this study [37].

### 2.6.3. The full coupling

Finally, the vascular networks and the Darcy domains were combined in a (nonlinear) system of equations $\mathbf{A}\mathbf{x} = \mathbf{b}$, $\mathbf{A} = \mathbf{A}(\mathbf{x})$, with unknown $\mathbf{x}$ consisting of the pressure and the flow rate in the model

$$\begin{pmatrix} \mathbf{A}_{N-N} & \mathbf{A}_{N-T} & 0 \\ \mathbf{A}_{T-N} & \mathbf{A}_{T-T} & \mathbf{A}_{T-D} \\ 0 & \mathbf{A}_{D-T} & \mathbf{A}_{D-D} \end{pmatrix} \begin{pmatrix} \mathbf{x}_N \\ \mathbf{x}_T \\ \mathbf{x}_D \end{pmatrix} = \begin{pmatrix} \mathbf{b}_N \\ 0 \\ 0 \end{pmatrix}. \tag{2.20}$$

The indexes $N$ and $T$ refer to the internal and terminal nodes in the vascular network, and index $D$ stands for the Darcy equation discretization. The unknown $\mathbf{x}_N$ is the pressure at the internal nodes and flow rate in the corresponding segment, $\mathbf{x}_T$ is the pressure in the terminal nodes, and $\mathbf{x}_D$ is the pressure on the Darcy domain. The blocks matrices represent systems of equations in the following domains:

— Matrix $\mathbf{A}_{N-N} \in \mathbf{R}^{N \times N}$ is a matrix representation of the vascular graph model's system of nonlinear equations. It is based on equations (2.3)–(2.5) and consists of as many equations as the number of vessels and nodes in the internal nodes network. It also has no more than four non-zero entries per row.
— Matrix $\mathbf{A}_{T-T} \in \mathbf{R}^{T \times T}$ is the system of nonlinear equations on terminal nodes at the coupling between the vascular graph and the capillary model. It is based on equations (2.3) and (2.5). The matrix is a diagonal matrix.
— Matrices $\mathbf{A}_{N-T} \in \mathbf{R}^{N \times T}$ and $\mathbf{A}_{T-N} \in \mathbf{R}^{T \times N}$ are couplings between the internal nodes and the terminal nodes from equations (2.3)–(2.5).
— Matrix $\mathbf{A}_{D-D} \in \mathbf{R}^{D \times D}$ is the system of linear equations based on the Darcy's Law discretization in equation (2.19). Its number of variables is the number of discretization cells in the capillary domain and the system is symmetric.
— Matrices $\mathbf{A}_{D-T} \in \mathbf{R}^{D \times T}$ and $\mathbf{A}_{T-D} \in \mathbf{R}^{T \times D}$ are couplings between the terminal nodes and Darcy discretization cells. The pressure and mass continuity are based on equations (2.13) and (2.12).
— The $\mathbf{b}_N$ term in the r.h.s. is from the boundary conditions of the circulation system in equation (2.14).

Despite the nonlinearity in the governing equations, it can be solved efficiently by using the Schur complement method. Let

$$\mathbf{B} = \begin{pmatrix} \mathbf{A}_{T-T} & \mathbf{A}_{T-D} \\ \mathbf{A}_{D-T} & \mathbf{A}_{D-D} \end{pmatrix}.$$

The Schur complement of block $\mathbf{A}_{D-D}$ for the matrix $\mathbf{B}$ is defined as $\mathbf{B}/\mathbf{A}_{D-D} := \mathbf{A}_{T-T} - \mathbf{A}_{T-D}\mathbf{A}_{D-D}^{-1}\mathbf{A}_{D-T}$. $\mathbf{A}_{D-D}^{-1}$ is not computed explicitly, due to of the large size of the matrix. Instead, the system of linear equation $\mathbf{A}_{D-D}\mathbf{C} = \mathbf{A}_{D-T}$ is solved giving $\mathbf{C} = \mathbf{A}_{D-D}^{-1}\mathbf{A}_{D-T}$. Substituting back in matrix $\mathbf{A}$ (2.20), we have

$$\begin{pmatrix} \mathbf{A}_{N-N} & \mathbf{A}_{N-T} \\ \mathbf{A}_{T-N} & \frac{\mathbf{B}}{\mathbf{A}_{D-D}} \end{pmatrix} \begin{pmatrix} \mathbf{x}_N \\ \mathbf{x}_T \end{pmatrix} = \begin{pmatrix} \mathbf{b}_N \\ 0 \end{pmatrix}. \tag{2.21}$$

The above nonlinear system (2.21) is solved using an iterative nonlinear least-square optimization algorithm based on the Powell dogleg method [38]. The solution of the linearized system (by omitting the nonlinearities) was used as the initial approximation. The entire framework was solved using MATLAB 2019b on a desktop PC with an Intel(R) Core(TM) i7-7700 CPU 3.60GHz and 32 GB RAM, and the codes and instructions for reproducing the results are included in the dataset [39].

## 2.7. Vascular structures in image data

The model described above is generic and applicable to any two-dimensional or three-dimensional domain. As an illustrative example, we used the model to simulate blood flow in a frog tongue from a two-dimensional anatomical image from a classical biology textbook [23]. This example resembles a realistic vascular system while still being more easy to visualize, interpret and analyse compared to a full three-dimensional system. A drawback of this choice is that the original sample is not available, thus we cannot validate the correctness of the vessel structure. Furthermore, there is an apparent deformation of the geometry. The vascular networks are simplified into two flat networks, as shown in figure 1 as we do not have information about thickness and curvature in the third dimension. Incorrect vessel structure and incorrect geometry may have an impact on the blood flow in the entire domain, especially for intertwined vessels, which are not captured from the two-dimensional image. Still, we believe that the balance between arteries and veins in the image is highly realistic. Since the application to this two-dimensional domain is included as an illustrative example, a deeper analysis of the limitations is outside the scope of the current work.

Figure 3 shows the original image of the frog tongue, with the physiologist's segmentation of the main vessels. The frog tongue was stretched flat and nailed to a canvas. Thereafter, the trained physiologist manually traced the arterial and venous network structures on it. One may notice that only a few vessels serve the middle part of the tongue.

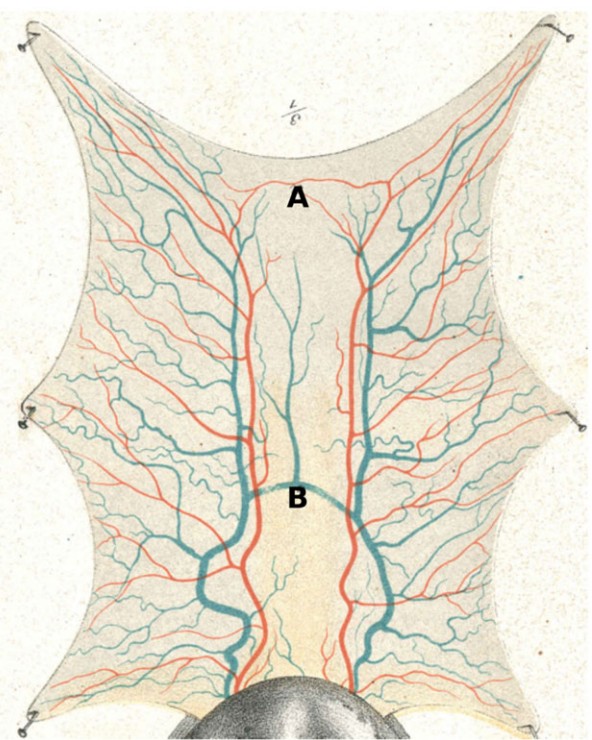

**Figure 3.** An anatomical frog tongue image from a classical textbook [23], with arterial (red) and venous (blue) vascular network structures. The networks are obtained by segmenting the anatomical vessel structures. Our capillary domain is the region inside the tongue's boundary. Vessel A connects two arterial networks, while Vessel B connects two vein networks.

**Table 1.** Vascular model parameters.

| parameter | value | unit | reference |
|---|---|---|---|
| capillary model size (two-dimensional) | $515 \times 634$ | pixel | — |
| real size | $30.9 \times 38$ | mm | [23] |
| porosity of the capillary bed ($\phi$) | 0.1 | dimensionless | [12] |
| permeability of arterial compartment ($K_a$) | $3 \times 10^{-6}$ | mm$^2$ | [41] |
| permeability of venous compartment ($K_v$) | $6 \times 10^{-6}$ | mm$^2$ | [41] |
| perfusion parameter ($\alpha$) | $5 \times 10^{-4}$ | kg$^{-1}$ mm s$^{-1}$ | [3] |
| viscosity of blood ($\mu$) | $3 \cdot 10^{-6}$ | kPa s$^{-1}$ | [3] |
| artery inlet pressure | 30 | mmHg | [22] |
| vein outlet pressure | 7.5 | mmHg | [24] |
| Young modulus ($E$) | 1 | MPa | [28] |
| Poisson's ratio ($\lambda$) | 0.5 | dimensionless | [28] |
| arterial vessel wall thickness ratio | 0.22 | dimensionless | [24] |
| venous vessel wall thickness ratio | 0.1 | dimensionless | [24] |

Both the arterial and venous network structures consist of two main vessels (left and right). Our network structure was based on an image segmentation of these using the method in [40]. The small arteries and veins that are near the roots but not visibly connected to the main vessels were discarded in our segmentation, as we need control over inlets and outlets.

We observed that both the arterial and venous networks have some level of anastomoses. The two arterial networks are connected in the uppermost part of the tongue (point A in figure 3) by a small vessel, and the venous networks are connected in the middle of the tongue by a large vessel (point

B). These connected structures have a regulation effect, maintaining blood circulation for the whole organ when some vessel is severed or occluded.

The frog tongue model parameters for our simulations are found in table 1. The real size of the image was estimated based on information from the original source [23]. The pressure inlet is defined as the blood pressure in an average adult frog *Pseudis paradoxsus* [22], and the pressure outlet is assumed to be equal to the average vein pressure in a male frog *Rana pipiens* mesentery [24]. The vessel elasticity parameters were chosen based on the mechanical properties of vessels around those sizes [21,27,28]. The isotropic and homogeneous permeability of the capillary bed was estimated using a simplified Darcy in scalar setting

$$\frac{Q}{A} = -\frac{K_\beta}{\mu}\frac{\Delta P}{\Delta L}, \quad \beta = a, v, \tag{2.22}$$

with frog cerebral cortex values as reference data [42]. The blood velocity is approximately 0.5 mm s$^{-1}$ in an arteriole with length of 2 mm for a pressure drop of $\Delta P = 7.5$ mmHg. We obtain an estimated permeability of $1.2 \times 10^{-6}$ mm$^2$. However, the value was adjusted in the simulation (refer to table 1) to match a normal blood flow in veins vessels based on the experimental data in *Rana pipiens*, which is $166.61 \pm 21.33$ µm s$^{-1}$ (mean ± standard deviation) in the veins with a radius around 10–20 µm [42]. The venous capillary compartment was assumed to be twice as permeable as the arterial capillary compartment.

# 3. Results and discussion

Using the model proposed above, we have simulated blood flow in several numerical experiments: a baseline model simulation, partial models and linearized simulations, and simulations with vessel occlusions. In particular, the occlusion experiments demonstrate possible applications to the study of blood circulation for an alteration or pathology of the vascular system.

## 3.1. Baseline model

The baseline model is the fully nonlinear model with nonlinearities due to both pressure drop at junctions as well as vessel elasticity. It was based on equations (2.1), (2.2), (2.4), (2.10), (2.5), (2.12) and (2.13). Figure 4 shows the computed pressure distribution in the whole vascular domain. The simulated pressure distribution has a good resemblance to the experimental data on *Rana pipiens* mesentery in [24]. We see a stronger agreement with the data for the big vessels compared to the smaller vessels close to the capillary bed (labels *a* and *b* in figure 4). The smaller vessels appear to have an overestimated pressure exceeding the observed data slope. This may be caused by rounding discretization effects as the vessel radius approaches pixel size. Further, the frog tongue vascular image from the textbook [23] was hand-drawn with a pen, with no emphasis on estimating diameters. In addition, the tongue was flat-stretched, thus deformed from its original shape. The inaccuracy of vessel radii caused by a combination of the three above-mentioned reasons may result in a lower pressure drop across the vessel compared to the anatomically realistic pressure drop (label a in figure 4). We might expect the flow pattern in the system to be impacted accordingly.

The total computational time for the baseline simulation is 82.6 s, with the nonlinear equation (2.21) taking 63.1 s to solve. The total pressure drop in the whole system was 22.5 mmHg (table 1), with the arterial network being the major contributor (71.6% of total). Further, the arterial compartment contributed with 2.7%, the venous compartment with 2.8%, and the vein vascular network had a 5.15 mmHg pressure drop (22.9%). These values are in a realistic range and indicate the importance of the arterial vascular structure to provide the blood circulation through the whole organ. If there is an alteration on an arterial vessel, its impact for the blood circulation will be greater than for a similar vein alteration. The pressure distribution is consistent with the distribution on humans or other animals [25,43,44]. From figure 4, we see that the arterioles are the biggest contributor to the pressure drop in the whole vasculature. The arterioles with a radius less than 100 µm provided 80% of the total pressure drop in the arterial network. In contrast to our result, the numerical experiments in [45] show that the smallest capillary vessels ($r < 5$ µm) contribute the most resistance in the mouse cortex micro-circulation simulation. Their haemodynamic modelling was conducted in a micro-scale vascular network with diameter less

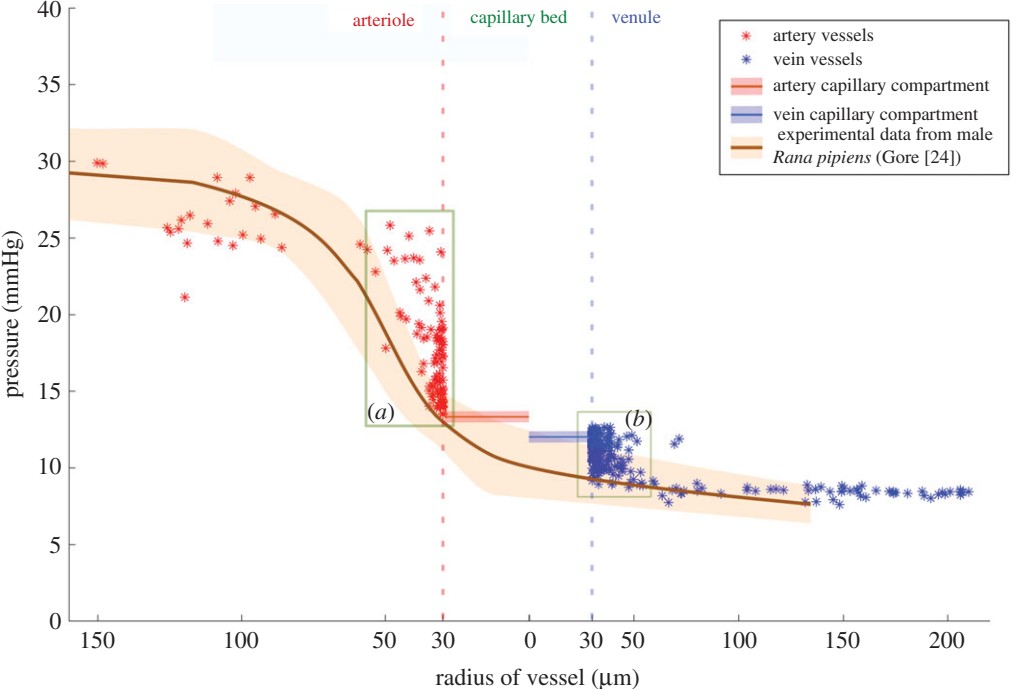

**Figure 4.** The static blood pressure distribution from our simulations has good estimation across the whole vascular system. The computed pressures (asterisks) follow the effective pressure from experimental data (brown solid line), the light brown area representing the variation from several experimental data measurements [24]. Our simulations produce some overestimation in the small arteriole (label *a*) and venule (label *b*) regions. The inaccurate simulation pressure is likely caused by vessel diameters in the range of the pixel size, so that the rounding of the vessel radius in the segmentation is in mismatch to the true value.

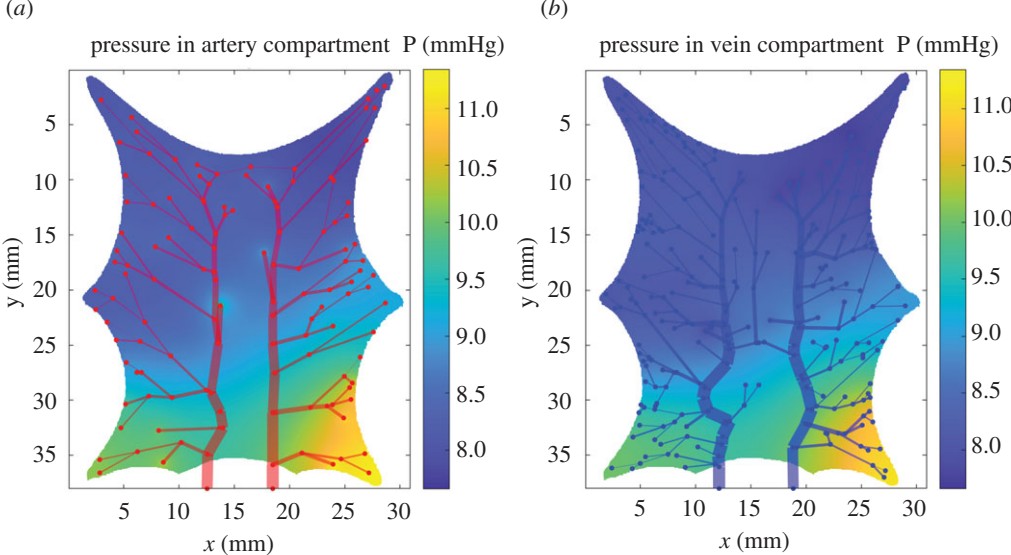

**Figure 5.** The pressure distribution for the baseline model: the segment thickness in the vascular graphs is proportional to the flow rate passing through it and the local pressure distribution in the Darcy domain. (*a*) The arterial network structure and the arterial compartment with pressure ranging from 7.6 to 11 mmHg (8.6 ± 0.7 mmHg). The right arterial vessel input provides 16% more blood flow compared to the left vessel, causing higher pressure in the right area. (*b*) The vein network structure and venous compartment with pressure ranging from 7.4 to 10.8 mmHg (8.5 ± 0.8 mmHg). The right vein vessel output has 4.3 times more blood flow than the left vessel, playing a vital role in blood distribution in the vein compartment.

than 50 µm, which, in our simulation set-up, is inside the range of the capillary bed (modelled as a Darcy domain). Hence, this particular detail could not be captured in our experiment due to the difference in the simulation scale and set-up.

**Table 2.** Numerical simulation results for several experiments: flow resistance, like the electrical resistance, is equal to the total pressure drop divided by the total volumetric flow in the system, $R = (P_{in} - P_{out})/Q$ (kg mm$^{-4}$ s$^{-1}$). The indexes $a_N$ and $v_N$ refer to the arterial and venous vascular networks, the indexes $a_C$ and $v_C$ to the arterial and vein capillary compartments respectively. The total computational time for simulation is denoted by $t_{tot}$, while $t_{sys}$ refers to the time for the solution of the system (2.21).

| simulation | $Q_{total}$ (mm$^3$ s$^{-1}$) | $R_{total}$ (kg mm$^{-4}$ s$^{-1}$) | $R_{a_N}$ | $R_{v_N}$ | $R_{a_C}$ | $R_{v_C}$ | $t_{tot}(t_{sys})$ (s) |
|---|---|---|---|---|---|---|---|
| baseline model[a] | 1.644 | 1.825 | 1.306 | 0.418 | 0.049 | 0.051 | 82.6 (63.1) |
| elasticity model[b] | 1.646 | 1.823 | 1.307 | 0.416 | 0.049 | 0.051 | 66.5 (47.1) |
| junction model[c] | 1.643 | 1.825 | 1.307 | 0.418 | 0.049 | 0.051 | 90.2 (70.7) |
| linear model | 1.754 | 1.710 | 1.319 | 0.357 | 0.018 | 0.017 | 19.4 ($10^{-5}$) |
| artery-1 occlusion[a] | 1.410 | 2.127 | 1.614 | 0.420 | 0.046 | 0.047 | 3216.4 |
| artery-2 occlusion[a] | 1.286 | 2.332 | 1.819 | 0.411 | 0.050 | 0.052 | 374.1 |
| vein-1 occlusion[a] | 1.644 | 1.824 | 1.297 | 0.442 | 0.042 | 0.044 | 757.2 |
| vein-2 occlusion[a] | 1.554 | 1.930 | 1.320 | 0.488 | 0.060 | 0.062 | 157.1 |
| artery-1 occlusion | 1.460 | 2.054 | 1.663 | 0.354 | 0.019 | 0.017 | 20.3 |
| artery-2 occlusion | 1.339 | 2.240 | 1.850 | 0.359 | 0.016 | 0.015 | 20.1 |
| vein-1 occlusion | 1.724 | 1.740 | 1.319 | 0.387 | 0.018 | 0.017 | 20.1 |
| vein-2 occlusion | 1.709 | 1.755 | 1.319 | 0.401 | 0.018 | 0.017 | 20.1 |

[a]*Baseline model* refers to the fully nonlinear model.
[b]*Elasticity model*: baseline model omitting the pressure drop at junctions equation (2.2).
[c]*Junction model*: baseline model omitting the elasticity equation (2.5). Occlusion: a 50% reduction of the original artery/vein radius.

In [17], the authors demonstrated a qualitatively comparable result to the experimental data using isotropic and homogeneous porous media as the capillary bed in the human cerebral cortex. Our result shows a similar agreement, that is, the dual Darcy compartment provided a good approximation as a replacement for small arterioles, venules and capillaries (see the capillary pressure drop in figure 4). The defined capillary bed covered not only capillaries but also unresolved vessels with diameter less than 60 µm, in which small arterioles and venules are covered. Figure 5 shows the local pressure distribution in the capillary bed. It was observed that the pressure is high around the top edges and low in the left bottom edges of the domain. This result is related to the vascular structure having a small number of branches from the arterial input vessel in the lower region. Pressures at the right area for both compartments are high compared to the left region, reflecting the blood distribution in the vascular network. The right arterial vessel input provides 16% more blood flow compared to the left vessel and the right vein vessel output has 4.3 times more blood flow compared to the left vessel.

Note that there is a good correlation between the pressures in the arterial and venous compartments, see figure 5, as the pressure follows the same distribution in both compartments. The pixel-wise pressure difference between the compartments is expected to be the driving force for local perfusion.

## 3.2. The partial models and the linearized model

The simulation with the linearized model follows the same set-up as the baseline model, but without the nonlinearities due to vessel elasticity and pressure drops at junctions. We have also tested a setup involving only one of the mentioned nonlinearities at the time (partial models). The macroscopic changes caused by each nonlinearity were compared across the entire system, including the total volumetric flow, total flow resistance and average resistance of the model components. Flow resistance, like an electrical resistance, is equal to the pressure drop divided by the volumetric flow in the system, $R = (P_{in} - P_{out})/Q$ (kg mm$^{-4}$ s$^{-1}$). The flow resistance of the model component is an approximation; for example, the pressure drop in the arterial network is the difference between the average pressure at the arterial network terminals and the input pressure. In this section, the models are defined as:

— The *Elasticity model*: baseline model omitting the pressure drop at junctions equation (2.2).
— The *Junction model*: baseline model omitting the elasticity equation (2.5).

— The *Linear model*: fully linear model omitting both equations (2.2) (pressure drop at junctions) and (2.5) (elasticity equation) from the baseline model. The linearized model coincides with the model considered in [3]. The solution of this model is used as the initial value for the nonlinear solver in the other models.

### 3.2.1. Elasticity model

The vessel's elasticity operates similarly to the capacitance in an electrical system. Thus, elasticity can maintain pressure constant by increasing or decreasing the vessel radii. However, this effect also gives additional resistance to blood flow, equivalent to impedance in electricity. Table 2 reports and compares the flow resistance in the system, which is equal to the total pressure drop divided by the total volumetric flow and is analogous to an electrical resistor.

The arterial vessel tends to expand if the pressure in the surrounding Darcy domain is lower than vessel inner pressure. On the other hand, the vein vessel tends to tighten and increase resistance. In our experiments, the elasticity model gives a total resistance of $1.823 \, \mathrm{kg \, mm^{-4} \, s^{-1}}$ (table 2), which is smaller than the baseline model. The elasticity model provided some small differences in arterial flow input and the resistance distribution compared to the baseline model. The computational time is less than the baseline, with the nonlinear system being solved in $47.1 \, \mathrm{s}$ (table 2). The elasticity model has a shorter running time because it only modifies the Poiseuille equation (2.1) by removing the junction term from the baseline model. The linear model solution is used as the initial value for the nonlinear solver.

### 3.2.2. Junction model

The pressure drop at the junctions increased the total resistance in the vascular structure, both in arteries and veins [20]. It is also causing the total blood flow in the whole system to decrease, since the volumetric blood flow, $Q$, changes according to the relation $\Delta P = RQ$. This bifurcation set-up is known as pressure matching at a junction node, which introduces nonlinearity in the flow model [34]. The resistance distribution in the whole system is $0.00069 \, \mathrm{kg \, mm^{-4} \, s^{-1}}$ bigger than the baseline model resistance (after rounding the values, the difference is lost in table 2). The junction model takes slightly longer time than the baseline model, mostly due to taking a longer time to achieve convergence in the nonlinear system.

The junction and the elasticity models only incorporate one of the nonlinear equations into the system at a time. Both simulations give results almost similar to the simulation with the baseline model. These results indicate that the effect of the junction pressure drop and the vessels elasticity do not stack up when combined. Indeed, the elasticity decreases the resistance built up by the junction pressure drop relaxing the vessel walls across the network. In this comparison setup, both nonlinearities play an essential role in describing blood regulation in the organ.

### 3.2.3. Linear model

This model is the fully linear setup based on equations (2.1), (2.4), (2.10), (2.12) and (2.13). The total resistance in this model was lower compared to the above models, thus allowing a larger volumetric blood flow (table 2). The arterial structure provided 77.1% of the total resistance and venous structure 20.9%, the Darcy compartments contributing only 1.0% and 1.0% of the total value. This is a notable change in resistance distribution compared to the partial models and the baseline model. The total resistance was reduced by 6.3% with respect to the baseline model. This result is consistent with another independent study [19], in which it was found that the average resistance underestimation using a fully linearized one-dimensional blood flow model was in the range 4.6–6.4% for human cerebral networks modelling. These findings indicate that, by removing all nonlinearities, both the total resistance and the resistance distribution of the system are altered (table 3) .

## 3.3. Occlusion in root vessels

In this section, we demonstrate how our model framework can be applied to understand the blood flow regulation due to pathology or network alterations. A series of simulations were performed by occluding one root vessel at a time, in each network. The occlusions were in a range between 5% and 100% of the original vessel radius (figure 6). When the occlusion reaches 80% of the initial radius value, the resistance becomes almost constant, for both arteries and veins, and for both baseline and linear models. The

Table 3. Comparison of results of occlusion experiments in the literature. We cite excerpts from the original text.

| object | method | artery occlusion | vein occlusion |
|---|---|---|---|
| One of the inlet or outlet vessels in a frog tongue (this work) | Simulation | The pressure above the occluded part of the structure was notably lower than the rest. | The existence of collateral circulation provided a new drain to maintain blood circulation after venous occlusions. The vascular structure played a vital role in the flow regulation. |
| One penetrating vessel in a human cortex model [2] | Simulation | 'The central region of reduced blood pressure forms a conical shape, with the area of the region getting smaller with depth until the deep layers exhibit very little pressure drop'. 'The drop in flow affects a larger volume of tissue, spreading out further and deeper than the drop in blood pressure'. 'The micro-infarct volume dependence on vessel diameter is observed'. | 'The results are similar to those for the arterioles, with conical pressure change regions, except that when occluding the venule there is a conical pressure increase as opposed to a decrease'. 'The drops in flow however are still diffuse across the voxel'. |
| A single vessel of penetrating arterioles or venules in a rat cortex [46] | Experiment | 'An occlusion of either a penetrating arteriole or venule generated severely hypoxic conditions in the acute period of 6 h post-occlusion'. 'The occlusion of a single penetrating arteriole leads to a highly localized, nominally cylindrical region of tissue infarction over a course of 7 d'. | 'The chronic result of occlusion to a penetrating venule is unreported and not readily predicted, as penetrating venules outnumber arterioles in rodent cortex and are highly collateralized on the pial surface'. 'Indeed, occlusion of a penetrating venule generated a microinfarction with notable similarity to that caused by occlusion of a penetrating arteriole'. |
| A single vessel of penetrating arterioles or venules in a mice cortex [47] | Experiment | 'The experimental data for both penetrating arterioles and venules showed a complete or near complete cessation of flow close to the occlusion, whereas our calculated values were small, but non-zero (figure 7d,f). This discrepancy originates from the assumed linear relation of flux and pressure, which ignores the propensity of red blood cells to stall at low pressure differences.' | |
| A single vessel of penetrating arterioles in the rat cortex [48] | Experiment | 'We found no evidence for active vasodilation in neighbouring arterioles in response to a penetrating arteriole occlusion'. 'We observed a slight, but not statistically significant, drop in both RBC speed and in RBC flow in neighbouring penetrating arterioles after the occlusion, indicating that blood flow to the area surrounding the occlusion was mildly decreased'. 'Uniformly over a 250 µm radius region around the occluded vessel, closely connected vessels did not dilate'. | |
| A gerbil superior sagittal sinus (SSS) of the rat cortex (the observation of the venous network alteration) [49] | Experiment | 'The angiography regularly revealed complex venous collateral pathways and venous flow reversal after SSS ligation, but demonstrated no significant differences in the diameter between that at pre-ligation and at 120 min post-SSS ligation'. 'The anatomical structure and an opening of the collateral pathways of the venous drainage system are closely related to microcirculatory alterations after venous occlusion'. | |
| An ascending venule and surface venule occlusion in the rat neocortex [50] | Experiment | 'The median RBC speed in measured capillaries located within 100 µm of the occluded ascending venule decreased to approximately 26% of the baseline value, and returned to baseline with increasing distance from the occluded vessel'. 'We observed dramatic changes in the routing of blood flow through the capillary bed after AV occlusion'. 'We found that capillaries up to three branches upstream from the targeted venule dilated after the dot as compared with sham experiments, with an average diameter increase of approximately 25%'. 'Collateral surface venules, when present, helped maintain normal blood flow after surface venule occlusions'. 'Topological architecture had a large role in determining blood flow changes'. | |

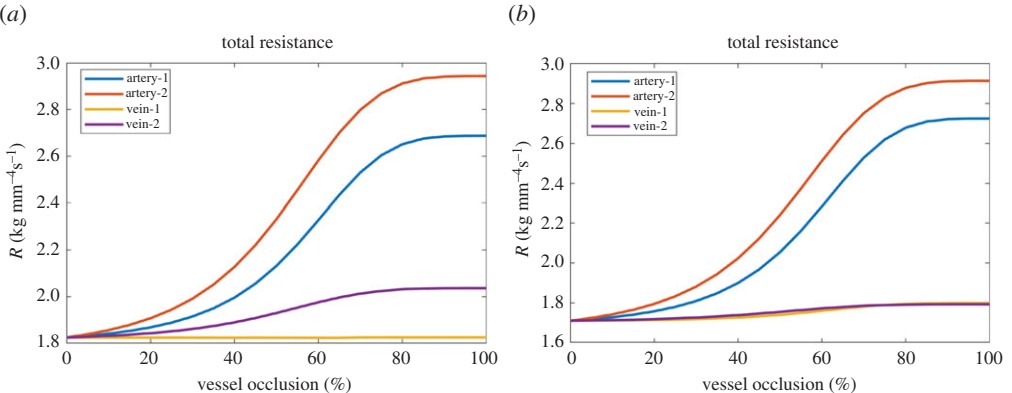

**Figure 6.** The vascular system resistance $R$ as a function of the occlusion rate in vessels: (*a*) Resistance estimates for occlusions, *baseline model*. (*b*) Resistance estimates for occlusions, *linear model*. The occlusion rate (*x*-axis) in a vessel varies from 5% to 100% of the original radius for each simulation. Arteries 1 and 2 refer to a segment in the right and left big vessel of the arterial structure. Veins 1 and 2 refer to a segment in the right and left big vessel of the venous structure. The artery occlusions cause an increment of the macroscopic flow resistance in the whole system for both the baseline and linear models. For the vein 1 occlusion, the entire vascular resistance is almost constant ($\pm$3% of the original total resistance). By contrast, the increment of the total resistance in vein 2 occlusion shows that the vein 2 plays vital role in draining blood from the organ tissue. This effect is only observed in the nonlinear simulation, while the linear model failed to capture this anomaly.

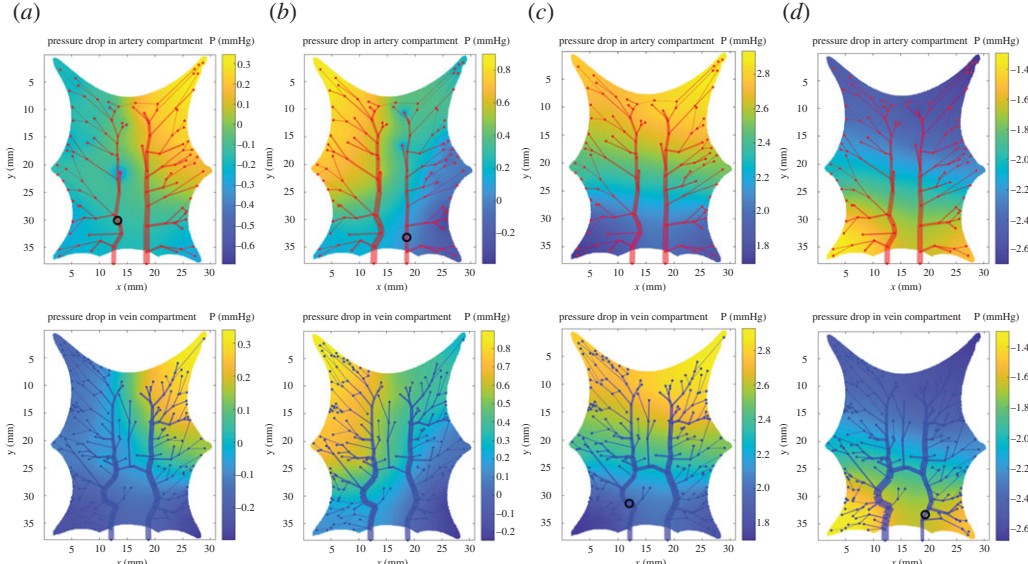

**Figure 7.** Pressure drop distribution in the capillary compartments with simulated occlusions (marked by black circle) computed with the baseline model. (*a,b*) Pressure drop distribution for artery occlusions are not mirroring each other due to the asymmetric structure of the arterial network with artery 1 occlusion generated bigger pressure drop. (*c*) Even with a negligible change in the macroscopic pressure drop (table 2), the pressure drop distribution in vein 1 experiment shows a notable pressure drop distribution for both compartments. (*d*) A higher pressure gradient between several terminal nodes in the bottom right corner enhances blood flow through the venous vessel rather than in the venous capillary. This microscopic alteration can occur with small change in the macroscopic flow.

occlusion experiments in the baseline model take considerably longer to run than the non-occluded experiments, due to the time required to achieve convergence in the nonlinear system (2.21). The occlusion location and the initial values determined using the linear model solution may be the cause of the large difference in convergence time (table 2).

### 3.3.1. Artery occlusion

Artery 1 and artery 2 occlusions generated expected results, increasing total resistances for both the occlusion on the baseline model and the linear model. The pressure above the occluded part of the

structure was notably lower than the rest. Furthermore, the artery 1 occlusion caused a steeper pressure gradient (left to right) compared to the artery 2 occlusion (figure 7a,b). This is due to the asymmetric structure of the left and right of the arterial network.

Table 2 reports a summary of the occlusion experiments with both baseline and linear models. Note that occlusions in the arterial roots increase resistance in the arterial structure and decrease total flow into the system. The driving pressure has to be raised around 20% compared to the non-occluded model to keep a similar volumetric blood flow. The artery 2 occlusion had a higher total resistance because the artery 2 vessel provides 16% more blood flow compared to the artery 1 vessel in the baseline model (figure 5). It shows that a prior higher blood flow in the occluded vessel will, in general, cause a more severe change to the system resistance. For artery occlusions, the linear model performs similarly to the baseline model, particularly for fully occluded vessels.

Figure 7 shows the pressure drop in the capillary compartments. Experimental studies of penetrating arteriole occlusion demonstrate similar effects. Table 3 summarizes the comparison between our findings and previous studies in the occlusion experiment. The observation in [46] reports a pressure drop in a region close to the occluded vessel and tissue hypoxia was detected in 6 h post-occlusion. The impact of vessel occlusion in penetrating arterioles in animal and human cortex has been studied both experimentally and numerically [2,46–48,51]. Although the two-dimensional frog tongue in this work differs extensively in terms of anatomical and functional properties to the cerebral cortex, the results show a good agreement for artery occlusion. In our simulation set-up, with only a pair of arteries and veins penetrating the capillary bed, the occlusion of one of the two main vessels had a more notable effect than for one vessel occlusion in the cerebral cortex, having instead several penetrating arterioles and venules [2,46].

### 3.3.2. Vein occlusion

The numerical results for the occluded venous networks show that the resistance changes for vein 1 and vein 2 occlusions had different character. While the vein 1 occlusion resulted in an almost constant total resistance, the vein 2 occlusion provided a resistance increase.

For the vein 1 occlusion, the total resistance only changed around ±3% of the baseline model, regardless of the occlusion percentage. This was due to vein 2 having 4.3 times more blood flow than vein 1, so that blood was drained from the vein compartment mainly through the vein 2 vessel. We also observed that the downstream vessels from the occlusion point were able to support blood influx from nearby terminals with an almost constant volumetric flow. In the converse situation, the vein 1 vessel could not support the blood flow when the vein 2 vessel was collapsed.

The vein 2 occlusion effects were consistent with venule occlusion in [2]. It generated a notable total resistance compared to the arteriole occlusion, except the capillary pressure increased instead of decreasing. In our simulation, the pressure drop in the vein 2 occlusion is less than for the artery occlusions (figure 7).

The simulated occlusions, both in arteries and veins, had a detectable impact to the blood circulation in general. These results are in good agreement with the experimental result on occlusions of one penetrating arterioles and venules in rats cortex (table 3) [46].

Without being necessarily physiologically consistent, the linear and baseline models occlusion results for vein 1 illustrate the models' capability to account for collateral circulation effects. A finer tuning of the modelling parameters may enable the system to maintain a close-to-constant net flow in varying anatomical setups. Yet, the present structure of the venous network has a good collateral circulation and maintains blood circulation in the whole system. This condition was mainly supported by the existence of anastomosis in the form of a big connected vessel in the venous network system, that allows flow across both sides of the vein network. These phenomena were consistent with the experimental result in [50], which showed that occlusion of one surface venule left the system's blood regulation almost unchanged when a collateral vessel provided a new drain.

Our vessel occlusion experiments do not take into account possible anatomical auto-regulating changes that can occur in the system over time. After a certain period, a real vascular system may change or heal to compensate for a blood circulation imbalance. This is a limitation of the model at present. An adaption of the model to allow for vessel tree growth would be highly interesting for instance in infarction modelling, but it is outside the main scope of this paper.

The resulting pressure distribution change in the venous compartment allows for blood going through the veins more than once and thereby also causes reduced blood flow in the capillary bed. Figure 7d shows an example of blood flow in the reverse direction due to a high-pressure difference

in the venous Darcy compartment. Higher pressure gradients between several terminal nodes in the bottom left corner make blood flow through venous vessels rather than in the capillary bed. In the vein 2 occlusion, around 5% of blood flow direction in the venous vessels was reversed, 28% of venous vessels had an average 23% blood flow increase, while the remaining vessels had a decrease of 16%. This result was also shown in the experimental result in the venule occlusion [50]: cortical ascending venules occlusions caused blood flow decrease and reversed flow direction (table 3). They also observed a radius dilation up to 25%, while the radii dilation in our simulation was less than 5%. The simulation design and parameters are crucial in determining vessel change, which may allow for greater radius dilation similar to experimental values.

# 4. Conclusion

We have introduced a nonlinear multi-scale modelling framework incorporating important features of blood flow in a full organ vasculature. The model is demonstrated using a two-dimensional geometry domain and vasculature from a frog tongue [23]. Our numerical experiments indicate that the modelling complexity is sufficient to account for important physiological and structural features. The combination of the explicit vessel representation and a generic Darcy model representation yields realistic full-organ simulations, without having a detailed knowledge of the local microvasculature. The framework is also highly flexible allowing for inclusion of other local nonlinear processes.

In the numerical experiments, the baseline simulations are in good agreement with empirical data from a male frog *Rana pipiens* mesentery [24]. Further, we conduct simulations with a pressure drop distribution in the whole vasculature consistent with the average distribution in other animals as well as humans [25,43,44]. The linearized simulations indicate the nonlinearities have an impact on the system. The simulations with partial models suggest that the nonlinear effects of vessel elasticity and pressure drop at vessel junctions do not stack up when combined, indicating that the two compensate for each other.

We illustrate the modelling potential by applying the method to vascular occlusions. Vessel networks alteration simulations confirm that both local and global blood circulations in the vascular system depend on the vascular structure. In our simulations, the existence of an anastomosis in form of a big connected vessel in the venous network system has an important role in maintaining blood circulation when a main root is occluded. We illustrate how an occlusion in the venous structure alters the microscopic flow resistance, while at the same time has negligible effects on the macroscopic flow.

Our simulations indicate that it is important to simulate the complete vascular system including the venous system, and not just the arterial system. In line with [50], our results of the venous occlusion experiments emphasize the importance of introducing elasticity and nonlinearities to provide a more accurate simulation of the blood flow.

Data accessibility. Data available on https://doi.org/10.5061/dryad.pg4f4qrn2.

Authors' contributions. All authors participated in the study design. U.Q. drafted the early version of the manuscript, and A.Z.M.K., J.M.N. and E.A.H. assessed and discussed the mathematical formulation and analysis of the methods. U.Q. drafted the figures. All authors edited the manuscript and approved the final version.

Competing interests. We declare we have no competing interests.

Funding. This work was supported by the Research Council of Norway under Frinatek 'Flow-based interpretation of dynamic contrast-enhanced MRI' project (NFR no. 262203).

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
