## [Peer Review File · Royal Society Open Science]

Review History

RSOS-201949.R0 (Original submission)

Review form: Reviewer 1

Is the manuscript scientifically sound in its present form?

Yes

Are the interpretations and conclusions justified by the results?

Yes

Is the language acceptable?

Yes

Do you have any ethical concerns with this paper?

No

Have you any concerns about statistical analyses in this paper?

No

Recommendation?

Major revision is needed (please make suggestions in comments)

Comments to the Author(s)

Abstract: The transition from the rather generic first sentence to the second sentence seems a bit abrupt to me. I would suggest to already focus more on blood flow/perfusion in the first sentence.

p2 l22 (and later): "et al." is not italicized consistently throughout the manuscript (cf. p4 l26). However, I would suggest dropping author names entirely, mentioning just a few authors but omitting most of them makes the reader wonder why this is the case.

p2 l38: The geometric simplification to straight cylinders between junctions is also common in other work, but to what extent does it impact the flow resistance? Even without taking into account curvature, some vessel segments become substantially shorter. Even if this probably cannot be investigated in detail in the present study, it would be interesting to have an estimate whether the impact of this simplification is negligible compared to the elasticity and junction terms considered or whether it can be expected to have substantial impact as well.

Fig 1 (and the surrounding text): Is my understanding correct that each of the two vascular systems is one connected network with two outflows as opposed to a tree? This is a strength compared to other work worth mentioning. But can it cause artifacts in the flow pattern or is this more of a possibility for checking results for plausibility?

p3 l50: should probably read "... viscoelasticity in the capillary bed"

p4 l38: "Vessel wall elasticity ... forces blood flow in a particular direction" is confusing to me, I would think that, vessels being elastic or not, the blood flows essentially in the direction along the vessel.

p4 l49: I would find it helpful to have references for each of the effects neglected in the model.

Eq 2.2, 2.4, 2.5, 2.14 and text/figure captions elsewhere: the formulas become easier to read if text abbreviations are not written as math (i.e., I would suggest using dat if I am not mistaken that the manuscript is written in latex).

p5 l49: How was the value of epsilon chosen, and does it somehow relate to the minimal vessel radius present in the vascular networks (i.e., the portion of what is actually vasculature, but represented as porous medium in the model)?

Eq 2.20: I find it confusing to write a non-linear system of equations as a matrix-vector product.

Fig 2: Extracting vascular networks from a 2D drawing is an interesting approach. Combined with Fig 1, am I correct that the two vascular networks are assumed to be (essentially) flat and separated instead of intertwined in three dimensions? That is a minor difference for the implementation of the flow model, but a large simplification from an anatomical point of view that should be discussed. Moreover, the image is an illustration of the concept that radii become thinner and not necessarily an accurate representation of radii, as mentioned in p9 l58. Given the fourth power of r in Eq 2.1, I would expect a relatively sensitivity of the pressure (and resulting perfusion) to variations/errors in radii, in particular if intertwined 3D vascular networks were used and positions of the end points also come into play.

Table 1: I think unit 1 instead of - could be slightly easier to read as "dimensionless". Without reading all the code in detail, I noticed that the perfusion parameter alpha is one order of magnitude larger in the code (Frog_parameter.m l10, with a typo in "Perdusion" in the comment).

p9 l33: should probably read "... an estimated permeability ..."

p9 l58: I would suggest elaborating on the impact of possibly inaccurate radii.

Fig 4: These plots are well done in terms of image quality and sequential color scale. They are readable even though I printed the pdf on a cheap printer, in grayscale, and two pages on one. Just a minor issue: capitalization of the plot titles is inconsistent in Fig 5 compared to Figs 4 and 6.

p11 l57: I find the terminology "elasticity model", "junction model", and "linear model" confusing. The former two sound like they are building blocks of the full model, while they actually refer to the full model excluding one building block. The latter uses a mathematical property instead of a term describing the content. Also, "linearized" suggests to me that the nonlinearities have been approximated by a non-zero linear term and not dropped entirely. Unfortunately, I do not have constructive idea how it could be phrased instead.

p11 l58: should this read "... coincides with ..."?

Table 2 and p14 l7: For readers not familiar with flow resistances, a brief explanation of the units would be helpful, an exponent of four tends to be unexpected.

p12 l50: I disagree with the conclusion that "[t]his experiment highlights the importance of inducing nonlinearities into the system.", or maybe I misunderstand it. In these in silico experiments, the additional terms certainly have an influence on the results, so being able to include non-linearities in modeling is important. However, without validation, it is not clear that the more complex model produces better results.

p13 l27f: What does "total resistance was reduced or arterial flow was overestimated" mean? For the "linear" model, the resistance is constant and solely determined by geometry, but not for the other models, right? So "resistance was reduced" applies when assuming constant flow, whereas "flow was overestimated" assumes constant pressure difference between inflow and outflow? Moreover, "arterial flow was overestimated" suggests a comparison to venous flow (and a violation of mass conservation), but this is not the point here, is it?

Occlusion models: That is an interesting in-silico experiment. Limitations of this model are probably that the occlusion of one artery or vein will somehow affect the overall blood circulation, in particular other supplies/drainages of the same organ, and that vascular anatomy might change over time to compensate for such a perfusion imbalance. I would suggest mentioning this explicitly in the paragraph starting in l8 on p16.

p14 l50: "evacuated" suggests to me that the system is influenced by an active experiment, this is probably not meant here.

p15 l38: should this read "... frog tongue ..."?

p15 l54: I would recommend not using the term "(in)significant" outside a statistical context to avoid potential confusion.

p15 l56: should probably read "... including the venous ... just the arterial ..."

p19: Ref 20 looks incomplete, Ref 29 contains redundant information

Data availability: The images representing the dataset look fine. I did not read the matlab code in all detail; besides the parameter mentioned above, I did not see anything implausible.

Conflict of interests statement: I suggested to cite work I have authored. Please check and decide for yourself which citations really should be added.

Technical note: the pdf I downloaded for review contained the text twice, once including references, one without. (If I had to guess, the first part is a pdf uploaded by the authors, the second one generated from the also submitted latex source by the submission system?) I only looked at the first part, pages and line numbers above refer to this version.

Review form: Reviewer 2

Is the manuscript scientifically sound in its present form?

No

Are the interpretations and conclusions justified by the results?

No

Is the language acceptable?

Yes

Do you have any ethical concerns with this paper?

No

Have you any concerns about statistical analyses in this paper?

No

Recommendation?

Major revision is needed (please make suggestions in comments)

Comments to the Author(s)

Comments to the authors are included in the attached report (see Appendix A).

Decision letter (RSOS-201949.R0)

Dear Mr Qohar

The Editors assigned to your paper RSOS-201949 "A nonlinear multi-scale model for blood circulation in a realistic vascular system" have now received comments from reviewers and would like you to revise the paper in accordance with the reviewer comments and any comments from the Editors. Please note this decision does not guarantee eventual acceptance.

Please submit your revised manuscript and required files (see below) no later than 21 days from today's (ie 09-Jun-2021) date. Note: the ScholarOne system will 'lock' if submission of the revision is attempted 21 or more days after the deadline. If you do not think you will be able to meet this deadline please contact the editorial office immediately.

on behalf of Dr Dirk Drasdo (Associate Editor) and Mark Chaplain (Subject Editor)
openscience@royalsociety.org

Associate Editor Comments to Author (Dr Dirk Drasdo):

Comments to the Author:

Dear Authors,

in the summary file, the pages 21ff seem to be an incomplete copy of the manuscript until page 20, with erroneous reference numbers and missing reference list.

It makes it easier for readers and reviewers to assess the novelties of a manuscript if those are explicitly stated (e.g. first time on frog tongue etc.).

You emphasise the computational efficiency of your approach compared to more detailed models. In that case it would be informative if the simulation times of both approaches for the same application could be estimated.

With kind regards
Dirk Drasdo

Reviewer comments to Author:

Reviewer: 1

Comments to the Author(s)

Abstract: The transition from the rather generic first sentence to the second sentence seems a bit abrupt to me. I would suggest to already focus more on blood flow/perfusion in the first sentence.

p2 122 (and later): "et al." is not italicized consistently throughout the manuscript (cf. p4 126). However, I would suggest dropping author names entirely, mentioning just a few authors but omitting most of them makes the reader wonder why this is the case.

p2 138: The geometric simplification to straight cylinders between junctions is also common in other work, but to what extent does it impact the flow resistance? Even without taking into account curvature, some vessel segments become substantially shorter. Even if this probably cannot be investigated in detail in the present study, it would be interesting to have an estimate whether the impact of this simplification is negligible compared to the elasticity and junction terms considered or whether it can be expected to have substantial impact as well.

Fig 1 (and the surrounding text): Is my understanding correct that each of the two vascular systems is one connected network with two outflows as opposed to a tree? This is a strength compared to other work worth mentioning. But can it cause artifacts in the flow pattern or is this more of a possibility for checking results for plausibility?

p3 150: should probably read "... viscoelasticity in the capillary bed"

p4 138: "Vessel wall elasticity ... forces blood flow in a particular direction" is confusing to me, I would think that, vessels being elastic or not, the blood flows essentially in the direction along the vessel.

p4 149: I would find it helpful to have references for each of the effects neglected in the model.

Eq 2.2, 2.4, 2.5, 2.14 and text/figure captions elsewhere: the formulas become easier to read if text abbreviations are not written as math (i.e., I would suggest using `\text{dat}` if I am not mistaken that the manuscript is written in latex).

p5 149: How was the value of epsilon chosen, and does it somehow relate to the minimal vessel radius present in the vascular networks (i.e., the portion of what is actually vasculature, but represented as porous medium in the model)?

Eq 2.20: I find it confusing to write a non-linear system of equations as a matrix-vector product.

Fig 2: Extracting vascular networks from a 2D drawing is an interesting approach. Combined with Fig 1, am I correct that the two vascular networks are assumed to be (essentially) flat and separated instead of intertwined in three dimensions? That is a minor difference for the implementation of the flow model, but a large simplification from an anatomical point of view that should be discussed. Moreover, the image is an illustration of the concept that radii become thinner and not necessarily an accurate representation of radii, as mentioned in p9 158. Given the fourth power of r in Eq 2.1, I would expect a relatively sensitivity of the pressure (and resulting perfusion) to variations/errors in radii, in particular if intertwined 3D vascular networks were used and positions of the end points also come into play.

Table 1: I think unit 1 instead of - could be slightly easier to read as "dimensionless". Without reading all the code in detail, I noticed that the perfusion parameter alpha is one order of magnitude larger in the code (Frog_parameter.m l10, with a typo in "Perdusion" in the comment).

p9 133: should probably read "... an estimated permeability ..."

p9 l58: I would suggest elaborating on the impact of possibly inaccurate radii.

Fig 4: These plots are well done in terms of image quality and sequential color scale. They are readable even though I printed the pdf on a cheap printer, in grayscale, and two pages on one. Just a minor issue: capitalization of the plot titles is inconsistent in Fig 5 compared to Figs 4 and 6.

p11 l57: I find the terminology "elasticity model", "junction model", and "linear model" confusing. The former two sound like they are building blocks of the full model, while they actually refer to the full model excluding one building block. The latter uses a mathematical property instead of a term describing the content. Also, "linearized" suggests to me that the nonlinearities have been approximated by a non-zero linear term and not dropped entirely. Unfortunately, I do not have constructive idea how it could be phrased instead.

p11 l58: should this read "... coincides with ..."?

Table 2 and p14 l7: For readers not familiar with flow resistances, a brief explanation of the units would be helpful, an exponent of four tends to be unexpected.

p12 l50: I disagree with the conclusion that "[t]his experiment highlights the importance of inducing nonlinearities into the system.", or maybe I misunderstand it. In these in silico experiments, the additional terms certainly have an influence on the results, so being able to include non-linearities in modeling is important. However, without validation, it is not clear that the more complex model produces better results.

p13 l27f: What does "total resistance was reduced or arterial flow was overestimated" mean? For the "linear" model, the resistance is constant and solely determined by geometry, but not for the other models, right? So "resistance was reduced" applies when assuming constant flow, whereas "flow was overestimated" assumes constant pressure difference between inflow and outflow? Moreover, "arterial flow was overestimated" suggests a comparison to venous flow (and a violation of mass conservation), but this is not the point here, is it?

Occlusion models: That is an interesting in-silico experiment. Limitations of this model are probably that the occlusion of one artery or vein will somehow affect the overall blood circulation, in particular other supplies/drainages of the same organ, and that vascular anatomy might change over time to compensate for such a perfusion imbalance. I would suggest mentioning this explicitly in the paragraph starting in l8 on p16.

p14 l50: "evacuated" suggests to me that the system is influenced by an active experiment, this is probably not meant here.

p15 l38: should this read "... frog tongue ..."?

p15 l54: I would recommend not using the term "(in)significant" outside a statistical context to avoid potential confusion.

p15 l56: should probably read "... including the venous ... just the arterial ..."

p19: Ref 20 looks incomplete, Ref 29 contains redundant information

Data availability: The images representing the dataset look fine. I did not read the matlab code in all detail; besides the parameter mentioned above, I did not see anything implausible.

Conflict of interests statement: I suggested to cite work I have authored. Please check and decide for yourself which citations really should be added.

Technical note: the pdf I downloaded for review contained the text twice, once including references, one without. (If I had to guess, the first part is a pdf uploaded by the authors, the second one generated from the also submitted latex source by the submission system?) I only looked at the first part, pages and line numbers above refer to this version.

Reviewer: 2

Comments to the Author(s)

Comments to the authors are included in the attached report.

===PREPARING YOUR MANUSCRIPT===

===PREPARING YOUR REVISION IN SCHOLARONE===

Author's Response to Decision Letter for (RSOS-201949.R0)

See Appendix B.

RSOS-201949.R1 (Revision)

Review form: Reviewer 1

Is the manuscript scientifically sound in its present form?

Yes

Are the interpretations and conclusions justified by the results?

Yes

Is the language acceptable?

Yes

Do you have any ethical concerns with this paper?

No

Have you any concerns about statistical analyses in this paper?

No

Recommendation?

Accept with minor revision (please list in comments)

Comments to the Author(s)

The authors have addressed all my comments in their revised submission. I have just a few minor points below, otherwise I recommend to accept the manuscript for publication.

In the page and approximate line numbers, I am referring to the version with highlighted changes (second one in the pdf I received from the submission system):

Title page: says "© 2014" (probably by the template and to be updated anyway)

p2 l30: "they" is unclear (it is neither the capillaries and arterioles nor the authors of [10]), you'll probably have to repeat "the authors"

Fig. 1: Red, blue, and green are indistinguishable in my greyscale printout. I would recommend using a different line style (e.g., dotted) for the green correspondence, as this is technically different from the red and blue ones (for which indistinguishable colors are not a problem).

p3 l54: should read "a homogeneous and isotropic porous medium" (not "an" and singular)

Fig. 2: I do not think this diagram is helpful, at least not in its present form. If I understand correctly, (2.1) is the formula for a single vessel segment, so three boxes with ΔP^h_n suggests that all three segments at a junction individually have the same pressure drop. Similarly, (2.2) is an additional pressure drop depending on the parent and a single offspring segment, so two boxes with ΔP^b_n on the left hand side suggest that the pressure drop is the same for both offspring segments. On the right hand side, why is there a single ΔP^b_n even though there are two datum vessels? It seems to me that more indices are needed here (dat, n₁, n₂ on the left; dat₁, dat₂, n on the right) and if my understanding of the

formula is correct, two $\Delta P^{\{b\}_{\dots}}$ are needed upstream of the bifurcation? Should (2.2) better be written as $\Delta P^{\{b\}_{\{dat, n\}}}$ then?

p6 l28: Should this read "user-chosen"?

Eq. 2.12: could be integrated in the sentence as "according to the relation" (no colon)

p8 l22: should read "non-zero entries" or "non-zeroes" (without "entries"), but I think the latter might be confusing

p8 l55: I would recommend somehow referring to the data availability statement rather than just stating "are available".

p9 l28f "... optimisation ... linearized system": mix of British and American English (please check entire manuscript, I did not pay attention everywhere)

Fig. 3: Does this reproduced figure require some kind of copyright statement (has probably expired)?

p10 l25 (and elsewhere in the manuscript): Should the species names be provided in parentheses?

p10 l37: Does $x \pm y$ mean (arithmetic) mean \pm standard deviation throughout the manuscript? If so, I would recommend stating so at the first occurrence.

p14 l45 (and in the caption of Table 2): flow resistance is a phenomenon, an electrical resistor is a physical component; I find it a bit confusing to call this an analogy. To me, the corresponding phenomenon is electrical resistance. Similarly, elasticity corresponds to capacitance.

p15 l35: The difference is about 4.5 orders of magnitude smaller than the value referred to, is this a meaningful difference due to the model, or a rounding effect due to the numerical implementation of the computations?

Table 3: Is everything written in this table cited from the respective papers, or are you partially describing the results in your own words? I would suggest clearly marking what was cited here to avoid ambiguity who "We" is. Also, for [46], the right column contains a sentence without verb.

p18 l56: I would recommend not using "significant" outside statistical analyses.

p20 l25: The manuscript without highlighted changes contains a stray "1pc" before the ethics statement.

The reference list contains a mix of title case and lower case journal names and a stray "." in [31].

Review form: Reviewer 2

Is the manuscript scientifically sound in its present form?

Yes

Are the interpretations and conclusions justified by the results?

Yes

Is the language acceptable?

Yes

Do you have any ethical concerns with this paper?

No

Have you any concerns about statistical analyses in this paper?

No

Recommendation?

Accept with minor revision (please list in comments)

Comments to the Author(s)

Many thanks to the authors for their responses to the reviews, and updates to the paper.

Thanks for taking into account my suggestion of describing the model with a pictorial illustration (Fig 2), it helps a bit. Nevertheless, the use of the n notation on all segments is not fully accurate. In my understanding, n denotes the different vessel segments. The pressure drops in the figure should then be denoted as $\Delta P_0, \Delta P_1, \dots$

In this regard, the notation r_0 in equation (2.5) is confusing, since it can be interpreted as r_n with $n=0$ (radius of the first vessel segment), instead of the vessel radius without pressure gradient (which applies to all vessel segments)

The updated notations in section 2(e) makes it clearer, especially subscript t for the terminal node. But I am a bit confused with Ω_{β} , which is defined as “the Darcy capillary volume” (Page 6 line 4, diff file) and as “the capillary compartment domain connected to the terminal node” shortly after (Page 6 line 36). These are the same (right ?), and if so it doesn’t need to be re-introduced (in different words).

On Table 2 : I quite like the new version of the table, especially the new computational time column. But are the numbers on that column for the occlusion models t_{tot} or t_{sys} (t_{tot} I guess ?) Could you elaborate on the huge difference in terms of computational time regarding occlusions (baseline model) ? Also I noticed that the results in Table 2 (and earlier in the text) changed. Why is that ? What changed in the model compared to the initial version of the paper ?

On Table 3 : This table is very helpful, thanks for adding it. The text in the Object column contains random spaces. Moreover, I would recommend using quotation marks when quoting text from a paper, separating parts of the text that are not contiguous in the original text or using [...]. For example when citing [47] : “We found no evidence for active vasodilation in neighboring arterioles in response to a penetrating arteriole occlusion”, “We observed a slight, but not statistically significant, drop in both RBC speed and in RBC flow in neighboring penetrating arterioles after the occlusion, indicating that blood flow to the area surrounding the occlusion was mildly decreased”, ...

Thanks for giving details and instructions regarding the provided codes. For what it’s worth, I usually add the link to the available code as a reference, so that I can cite this reference in the text (might be useful Page 8, line 56, diff file).

Some more minor comments / typos :

- Page 4 line 33 (diff file) : refer <- refers
- Page 6 line 29 : can be a set to <- can be set to
- Page 7, line 7 : For each segment i <- For each segment n
- Page 19, line 8 : had an detectable impact <- had a detectable impact

Decision letter (RSOS-201949.R1)

Dear Mr Qohar

On behalf of the Editors, we are pleased to inform you that your Manuscript RSOS-201949.R1 "A nonlinear multi-scale model for blood circulation in a realistic vascular system" has been accepted for publication in Royal Society Open Science subject to minor revision in accordance with the referees' reports. Please find the referees' comments along with any feedback from the Editors below my signature.

Please submit your revised manuscript and required files (see below) no later than 7 days from today's (ie 04-Oct-2021) date. Note: the ScholarOne system will 'lock' if submission of the revision is attempted 7 or more days after the deadline. If you do not think you will be able to meet this deadline please contact the editorial office immediately.

on behalf of Dr Dirk Drasdo (Associate Editor) and Mark Chaplain (Subject Editor)
openscience@royalsociety.org

Reviewer comments to Author:

Reviewer: 1

Comments to the Author(s)

The authors have addressed all my comments in their revised submission. I have just a few minor points below, otherwise I recommend to accept the manuscript for publication.

In the page and approximate line numbers, I am referring to the version with highlighted changes (second one in the pdf I received from the submission system):

Title page: says "© 2014" (probably by the template and to be updated anyway)

p2 l30: "they" is unclear (it is neither the capillaries and arterioles nor the authors of [10]), you'll probably have to repeat "the authors"

Fig. 1: Red, blue, and green are indistinguishable in my greyscale printout. I would recommend using a different line style (e.g., dotted) for the green correspondence, as this is technically different from the red and blue ones (for which indistinguishable colors are not a problem).

p3 l54: should read "a homogeneous and isotropic porous medium" (not "an" and singular)

Fig. 2: I do not think this diagram is helpful, at least not in its present form. If I understand correctly, (2.1) is the formula for a single vessel segment, so three boxes with ΔP^h_n suggests that all three segments at a junction individually have the same pressure drop. Similarly, (2.2) is an additional pressure drop depending on the parent and a single offspring segment, so two boxes with ΔP^b_n on the left hand side suggest that the pressure drop is the same for both offspring segments. On the right hand side, why is there a single ΔP^b_n even though there are two datum vessels? It seems to me that more indices are needed here (dat, n₁, n₂ on the left; dat₁, dat₂, n on the right) and if my understanding of the formula is correct, two $\Delta P^b_{\{...\}}$ are needed upstream of the bifurcation? Should (2.2) better be written as $\Delta P^b_{\{dat, n\}}$ then?

p6 l28: Should this read "user-chosen"?

Eq. 2.12: could be integrated in the sentence as "according to the relation" (no colon)

p8 l22: should read "non-zero entries" or "non-zeroes" (without "entries"), but I think the latter might be confusing

p8 l55: I would recommend somehow referring to the data availability statement rather than just stating "are available".

p9 l28f "... optimisation ... linearized system": mix of British and American English (please check entire manuscript, I did not pay attention everywhere)

Fig. 3: Does this reproduced figure require some kind of copyright statement (has probably expired)?

p10 l25 (and elsewhere in the manuscript): Should the species names be provided in parentheses?

p10 l37: Does $x \pm y$ mean (arithmetic) mean \pm standard deviation throughout the manuscript? If so, I would recommend stating so at the first occurrence.

p14 l45 (and in the caption of Table 2): flow resistance is a phenomenon, an electrical resistor is a physical component; I find it a bit confusing to call this an analogy. To me, the corresponding phenomenon is electrical resistance. Similarly, elasticity corresponds to capacitance.

p15 l35: The difference is about 4.5 orders of magnitude smaller than the value referred to, is this a meaningful difference due to the model, or a rounding effect due to the numerical implementation of the computations?

Table 3: Is everything written in this table cited from the respective papers, or are you partially describing the results in your own words? I would suggest clearly marking what was cited here to avoid ambiguity who "We" is. Also, for [46], the right column contains a sentence without verb.

p18 l56: I would recommend not using "significant" outside statistical analyses.

p20 l25: The manuscript without highlighted changes contains a stray "1pc" before the ethics statement.

The reference list contains a mix of title case and lower case journal names and a stray "." in [31].

Reviewer: 2

Comments to the Author(s)

Many thanks to the authors for their responses to the reviews, and updates to the paper.

Thanks for taking into account my suggestion of describing the model with a pictorial illustration (Fig 2), it helps a bit. Nevertheless, the use of the n notation on all segments is not fully accurate. In my understanding, n denotes the different vessel segments. The pressure drops in the figure should then be denoted as $\Delta P_0, \Delta P_1, \dots$

In this regard, the notation r_0 in equation (2.5) is confusing, since it can be interpreted as r_n with $n=0$ (radius of the first vessel segment), instead of the vessel radius without pressure gradient (which applies to all vessel segments)

The updated notations in section 2(e) makes it clearer, especially subscript t for the terminal node. But I am a bit confused with Ω_{β} , which is defined as "the Darcy capillary volume" (Page 6 line 4, diff file) and as "the capillary compartment domain connected to the terminal node" shortly after (Page 6 line 36). These are the same (right ?), and if so it doesn't need to be re-introduced (in different words).

On Table 2 : I quite like the new version of the table, especially the new computational time column. But are the numbers on that column for the occlusion models t_{tot} or t_{sys} (t_{tot} I guess ?) Could you elaborate on the huge difference in terms of computational time regarding occlusions (baseline model) ? Also I noticed that the results in Table 2 (and earlier in the text) changed. Why is that ? What changed in the model compared to the initial version of the paper ?

On Table 3 : This table is very helpful, thanks for adding it. The text in the Object column contains random spaces. Moreover, I would recommend using quotation marks when quoting text from a paper, separating parts of the text that are not contiguous in the original text or using [...]. For example when citing [47] : "We found no evidence for active vasodilation in neighboring arterioles in response to a penetrating arteriole occlusion", "We observed a slight, but not statistically significant, drop in both RBC speed and in RBC flow in neighboring penetrating arterioles after the occlusion, indicating that blood flow to the area surrounding the occlusion was mildly decreased", ...

Thanks for giving details and instructions regarding the provided codes. For what it's worth, I usually add the link to the available code as a reference, so that I can cite this reference in the text (might be useful Page 8, line 56, diff file).

Some more minor comments / typos :

- Page 4 line 33 (diff file) : refer <- refers
- Page 6 line 29 : can be a set to <- can be set to
- Page 7, line 7 : For each segment i <- For each segment n
- Page 19, line 8 : had an detectable impact <- had a detectable impact

===PREPARING YOUR MANUSCRIPT===

===PREPARING YOUR REVISION IN SCHOLARONE===

- An individual file of each figure (EPS or print-quality PDF preferred [either format should be produced directly from original creation package], or original software format).
- An editable file of each table (.doc, .docx, .xls, .xlsx, or .csv).
- An editable file of all figure and table captions.

- Any electronic supplementary material (ESM).
- If you are requesting a discretionary waiver for the article processing charge, the waiver form must be included at this step.
- If you are providing image files for potential cover images, please upload these at this step, and inform the editorial office you have done so. You must hold the copyright to any image provided.
- A copy of your point-by-point response to referees and Editors. This will expedite the preparation of your proof.

- Ensure that your data access statement meets the requirements at <https://royalsociety.org/journals/authors/author-guidelines/#data>. You should ensure that you cite the dataset in your reference list. If you have deposited data etc in the Dryad repository, please only include the 'For publication' link at this stage. You should remove the 'For review' link.
- If you are requesting an article processing charge waiver, you must select the relevant waiver option (if requesting a discretionary waiver, the form should have been uploaded at Step 3 'File upload' above).
- If you have uploaded ESM files, please ensure you follow the guidance at <https://royalsociety.org/journals/authors/author-guidelines/#supplementary-material> to include a suitable title and informative caption. An example of appropriate titling and captioning may be found at https://figshare.com/articles/Table_S2_from_Is_there_a_trade-off_between_peak_performance_and_performance_breadth_across_temperatures_for_aerobic_scope_in_teleost_fishes_/3843624.

Author's Response to Decision Letter for (RSOS-201949.R1)

See Appendix C.

Decision letter (RSOS-201949.R2)

Dear Mr Qohar,

I am pleased to inform you that your manuscript entitled "A nonlinear multi-scale model for blood circulation in a realistic vascular system" is now accepted for publication in Royal Society Open Science.

on behalf of Dr Dirk Drasdo (Associate Editor) and Mark Chaplain (Subject Editor)
openscience@royalsociety.org

Appendix A

Review --- A non-linear multi-scale model for blood circulation in a realistic vascular system

This paper introduces a multi-scale model for blood circulation on a realistic vascular network, illustrated by simulations on a 2D frog tongue. The main novel contribution is the pressure drop at bifurcations and the dependence of vessel radius on pressure (elasticity) providing a good balance between complexity/accuracy and computational cost although inducing nonlinearities. The paper reports promising results for occlusions in root vessels, illustrating the potential of the model in such applications. Nevertheless, I believe efforts need to be made both regarding the description of the model and its validation, as detailed in the following comments.

Materials and methods

1. Figure 1 nicely illustrates the coupling between the components of the model but the (too many) dotted lines are a bit confusing. Even if it is quite obvious, the green dotted lines aren't introduced anywhere.
2. The description of the graph structure model involves quite a lot of notations. I recommend adding a pictorial illustration of the graph to make it clearer.
3. The results section compares the changes due to each nonlinearity. I believe the description of the model should match this, with the description of the linear model followed by a paragraph for each nonlinearity (from linear to baseline model).
4. (page 4, line 45) It is not clear what P_{in} and P_{ext} are. Are those given values? The pressure on the arterial and venous root nodes is also denoted as P_{ext} (2.14), is that the same P_{ext} ? Moreover, (2.14) seems to indicate that the pressure imposed at the arterial roots and venous roots is the same, while Table 1 states two values for artery inlet pressure (30 mmHg) and vein outlet pressure (7.5 mmHg) which looks more coherent.
5. Regarding Table 1, artery inlet pressure and vein outlet pressure are reported as coming from [22] while the text says pressure outlet in veins comes from [24] (page 9, line 25)
6. In (2.10), Ω_a and Ω_c are not introduced
7. The paragraph on coupling vascular structure and capillary models is very unclear (whereas crucial for the model):
 - a. The finite radius ϵ used with the distribution function is not properly defined. The notation N_ϵ as the number of cells in this radius seems to indicate that it represents a region of radius ϵ around the terminal node? How is it chosen?
 - b. The integration domain Ω in (2.12) is not properly defined
 - c. Notations aren't consistent: terminal node are denoted i while I is used in (2.13)
 - d. The short explanation on tube radius computation is a bit confusing, the notation r_t and r_c aren't used anywhere else in the paper.
8. (Page 6 line 19) The graph segment is denoted i while it was previously denoted n (on page 4).
9. (Page 6 line 54) Matrix A_D should be $A_{\{D-D\}}$ according to notations in the full coupling paragraph (Page 7)

10. (Page 7) Authors state that TPFA method is consistent and convergent on the quadrilateral grid used in this study. I recommend introducing the use of this quadrilateral grid earlier.
11. (Page 7, full coupling paragraph) The matrices $A_{\{D-T\}}$ and $A_{\{T-D\}}$ should also be listed, and the b_N term in the RHS is not introduced. The notation F representing the system is confusing : $F(x_n) = b_n$ represents the vascular graph model while $F(x) = b$ represents the full system. Moreover I don't think these F notations are really needed (but if so, these should be properly defined earlier when introducing the model).
12. (Figure 2) The point D mentioned on Page 9 line 18 is not visible on the figure (even if one can guess where it is).
13. (Page 9 line 34) "it was also manually adjusted". I guess it has to do with the permeability. Which value is set then ? And how different this is compared to the estimated permeability ?
14. There is no information about the software/solver used. The code and data is accessible but there is no instructions on how to run the model and reproduce the results.

Result and discussion

1. The impact of each of the nonlinearities added to the model (pressure drop at junctions and vessel elasticity) is an important contribution of the paper. Nevertheless, the comparison of the linear, elasticity, junction and baseline models remains a bit short. Table 2 clearly shows that both non-linearities have an essential role, but the text only describes the content of Table 2. It would be interesting to elaborate further on what (and where) are the differences between the results obtained from junction, elasticity and baseline models.
2. The presented results are compared to both modeling and experimental papers, with applications on human cerebral cortex or rats cortex. These comparisons are mentioned all through the result and discussion section, making it hard to follow. Adding a new table summarizing the comparisons that have been made (giving what's in good agreement and what are the differences) would help the reader to have a better understanding of the potential applications of this model and its limitations.
3. The application on occlusion in root vessels gives consistent results and shows again the importance of the nonlinearities. Nevertheless, the radius dilation effect is far from experimental findings (Page 15 line 27). This could be further detailed.

Minor comments / typos :

- Page 8 line 58 : Misplaced comma
- Page 9 line 6/7 : of the # limitations
- Page 14 line 41 : the blood regulation **is** highly dependent
- Page 15 line 29 : to allow**s**

Appendix B

Dear editor

Please find enclosed the revised version of the manuscript "A nonlinear multi-scale model for blood circulation in a realistic vascular system". We wish to thank the Referees for a careful reading of the article and for relevant and constructive comments. We have addressed all the comments and concerns to the best of our knowledge. Please refer to the text below for the Editor and Referee comments (in black) in *italic* and our author response (in blue). We hope that the Referee are satisfied with the changes and they find this revised version, suitable for publication in Royal Society Open Science.

Response to the Editor

in the summary file, the pages 21ff seem to be an incomplete copy of the manuscript until page 20, with erroneous reference numbers and missing reference list.

The incomplete copy with missing references is caused by the compiling error of the system. Hence, I included a pdf, as recommended in file submission instructions, for reference.

It makes it easier for readers and reviewers to assess the novelties of a manuscript if those are explicitly stated (e.g. first time on frog tongue etc.). You emphasise the computational efficiency of your approach compared to more detailed models. In that case it would be informative if the simulation times of both approaches for the same application could be estimated.

Thank you

Response to the reviewers

We thank the reviewers for their time spent carefully reviewing the manuscript, and in their opinions regarding the science and presentation of the material. The revision of the manuscript are listed according to the reviewer comments.

Reviewer #1

Comments to the Author(s)

1. *Abstract: The transition from the rather generic first sentence to the second sentence seems a bit abrupt to me. I would suggest to already focus more on blood flow/perfusion in the first sentence.*

We appreciate the reviewer's input and revise the first sentence to emphasize blood flow. The revised sentence is as follows: "In the last decade, numerical models have become an increasingly important tool in biological and medical science. Numerical simulations contribute to a deeper understanding of physiology and are a powerful tool for better diagnostics and treatment."

2. *p2 l22 (and later): "et al." is not italicized consistently throughout the manuscript (cf. p4 l26). However, I would suggest dropping author names entirely, mentioning just a few authors but omitting most of them makes the reader wonder why this is the case.*

Done, thanks.

3. *p2 l38: The geometric simplification to straight cylinders between junctions is also common in other work, but to what extent does it impact the flow resistance? Even without taking into account curvature, some vessel segments become substantially shorter. Even if this probably cannot be investigated in detail in the present study, it would be interesting to have an estimate whether the impact of this simplification is negligible compared to the elasticity and junction terms considered or whether it can be expected to have substantial impact as well.*

Indeed, this simplification could be an intriguing subject for further investigation, but it is outside of the primary scope of this study. We estimate that the effect is negligible for elasticity. This is because the vessel resistance is linearly proportional to the length, $R \approx L$, according to Poiseuille's law (2.1), while vessel resistance is inversely proportional to the radius of the vessel to the fourth power, $R \approx r^{-4}$. The junction term is proportional to the square of the blood flow rate, which

varies according to vessel size. Therefore we also expect the effect to be negligible. In further applications, the impact of curvature and vessel length can be further reduced using segmentation of higher resolution image data, provided these are available.

4. *Fig 1 (and the surrounding text): Is my understanding correct that each of the two vascular systems is one connected network with two outflows as opposed to a tree? This is a strength compared to other work worth mentioning. But can it cause artifacts in the flow pattern or is this more of a possibility for checking results for plausibility?*

The inflow and outflow of this network are from two inlets and outlets at the roots of the trees. We do not exclude the presence of possible artifacts, however we would expect such artifacts to depend on the location of the tree terminals rather than of the inlets and outlets. Again, this would depend on the imaging data available for the task.

5. *p3 l50: should probably read "... viscoelasticity in the capillary bed"*

Thank you very much. The error has been corrected.

6. *p4 l38: "Vessel wall elasticity ... forces blood flow in a particular direction" is confusing to me, I would think that, vessels being elastic or not, the blood flows essentially in the direction along the vessel.*

Thank you very much for bringing this to our attention. The imprecision has been corrected, and the revised sentence is as follows: "Vessel wall elasticity provides capacitance and pulse-wave dampening, allowing the arteries to maintain a relatively constant pressure despite the pulsating nature of the blood flow."

7. *p4 l49: I would find it helpful to have references for each of the effects neglected in the model.*

We have added the references for the effects neglected in the model (p3 l49).

8. *Eq 2.2, 2.4, 2.5, 2.14 and text/figure captions elsewhere: the formulas become easier to read if text abbreviations are not written as math (i.e., I would suggest using `textdat` if I am not mistaken that the manuscript is written in latex).*

Revised as suggested.

9. *p5 l49: How was the value of epsilon chosen, and does it somehow relate to the minimal vessel radius present in the vascular networks (i.e., the portion of what is actually vasculature, but represented as porous medium in the model)?*

We have added the following explanation: "The finite radius ϵ can be a set to a user chosen constant or to a fraction of the terminal vessel's radius. In this manuscript, it is set to three pixels to represent the vessels that are not observable in the imaging data. "

10. *Eq 2.20: I find it confusing to write a non-linear system of equations as a matrix-vector product.*

Although we understand that it might be confusing, we still decided to keep the matrix-vector product as a simple and schematic representation for the model coupling. It also has to do with the way the nonlinear system is solved. The initial solution for the nonlinear solver is the solution of the linear model (removing the nonlinearities altogether).

11. *Fig 2: Extracting vascular networks from a 2D drawing is an interesting approach. Combined with Fig 1, am I correct that the two vascular networks are assumed to be (essentially) flat and separated instead of intertwined in three dimensions? That is a minor difference for the implementation of the flow model, but a large simplification from an anatomical point of view that should be discussed. Moreover, the image is an illustration of the concept that radii become thinner and not necessarily an accurate representation of radii, as mentioned in p9 l58. Given the fourth power of r in Eq 2.1, I would expect a relatively sensitivity of the pressure (and resulting perfusion) to variations/errors in radii, in particular if intertwined 3D vascular networks were used and positions of the end points also come into play.*

We appreciate the reviewer's viewpoint and agree on the shortcomings of this approach. We are unable to extract a true anatomical vascular network due to data limitations. This simplification

may result in an incorrect calculation of the true pressure. In the Results section, we elaborated on this problem in the following sentences: "This may be caused by the discretization effects as we approached a region with vessel radius in the range of one single pixel. Thus, rounding the vessel radius in the segmentation can cause a mismatch to its true value in small vessels. Further, the frog tongue vascular image from the textbook [?] was hand-drawn with a pen, with no emphasis on estimating diameters in addition to the flat-stretched tongue shape. The inaccuracy of vessel radii caused by a combination of the three mentioned reasons may result in a lower pressure drop across the vessel compared to the anatomically realistic pressure drop (label a in figure ??). Further, the flow pattern in the system will be impacted accordingly."

12. *Table 1: I think unit 1 instead of - could be slightly easier to read as "dimensionless". Without reading all the code in detail, I noticed that the perfusion parameter alpha is one order of magnitude larger in the code (Frog parameter.m l10, with a typo in "Perdusion" in the comment).*

We thank the reviewer for pointing this out. We have revised accordingly.

13. *p9 l33: should probably read "... an estimated permeability ..."*

Corrected.

14. *p9 l58: I would suggest elaborating on the impact of possibly inaccurate radii.*

We thank the reviewer's suggestion and agree to elaborate the impact of of possibly inaccurate radii.

15. *Fig 4: These plots are well done in terms of image quality and sequential color scale. They are readable even though I printed the pdf on a cheap printer, in grayscale, and two pages on one. Just a minor issue: capitalization of the plot titles is inconsistent in Fig 5 compared to Figs 4 and 6.*

Thanks for pointing this out. We have revised accordingly.

16. *p11 l57: I find the terminology "elasticity model", "junction model", and "linear model" confusing. The former two sound like they are building blocks of the full model, while they actually refer to the full model excluding one building block. The latter uses a mathematical property instead of a term describing the content. Also, "linearized" suggests to me that the nonlinearities have been approximated by a non-zero linear term and not dropped entirely. Unfortunately, I do not have constructive idea how it could be phrased instead.*

We are aware that the the terminology could be confusing. We had a lengthy discussion about the choice of the terms. The reason why we decided to call the "full" model as "baseline" instead, is that a model can never be full enough, because one can always add extra effects. We decided to use "baseline" because we believe this is a good model to start with. It has enough nonlinearity so that it gives reasonable and realistic results in agreement with experimental findings, and it can be further enhanced by considering more complicated elements (like replacing the flow in the tree networks with CFD simulations, etc.) The choice of "elasticity" and "junction" were due to our interest in observing which of the two factors had most impact. In the baseline model, it appears that the effects compensate each other, so that none of them is dominating.

17. *p11 l58: should this read "... coincides with ..."?*

Corrected, thanks.

18. *Table 2 and p14 l7: For readers not familiar with flow resistances, a brief explanation of the units would be helpful, an exponent of four tends to be unexpected.*

We have revised and added the following explanation: "... flow resistance in the system, which is equal to the total pressure drop divided by the total volumetric flow and is analogous to an electrical resistor, $R = (P_{in} - P_{out})/Q$ [kg.mm⁻⁴s⁻¹]. ...". Since P [kg.mm⁻¹s⁻²] and Q [mm³s⁻¹], $\Delta P/Q$ will have an exponent of four in mm.

19. *p12 l50: I disagree with the conclusion that "[t]his experiment highlights the importance of inducing nonlinearities into the system.", or maybe I misunderstand it. In these in silico experiments, the additional terms certainly have an influence on the results, so being able to include non-linearities*

in modeling is important. However, without validation, it is not clear that the more complex model produces better results.

Because of the nature of the configuration and system, our simulations could not be validated experimentally. In addition, experimental data in frog mesentery are insufficient for validation of our 2D frog tongue simulation. We have removed the sentence altogether.

20. *p13 l27f: What does "total resistance was reduced or arterial flow was overestimated" mean? For the "linear" model, the resistance is constant and solely determined by geometry, but not for the other models, right? So "resistance was reduced" applies when assuming constant flow, whereas "flow was overestimated" assumes constant pressure difference between inflow and outflow? Moreover, "arterial flow was overestimated" suggests a comparison to venous flow (and a violation of mass conservation), but this is not the point here, is it?*

In this result, we compare the linear model to the baseline, which is thought to be closer to the true value. To compare the results according to the reference, we used a misleading term, "overestimation flow." Instead, we have revised to emphasize the "reduced flow resistance" in the linear model.

21. *Occlusion models: That is an interesting in-silico experiment. Limitations of this model are probably that the occlusion of one artery or vein will somehow affect the overall blood circulation, in particular other supplies/drainages of the same organ, and that vascular anatomy might change over time to compensate for such a perfusion imbalance. I would suggest mentioning this explicitly in the paragraph starting in l8 on p16.*

As the referee points out, at present our present model does not account for vascular changes over time and is assumed to be in a steady state condition. We have tested techniques for vessel growth but our results are far too premature to be included in this work. We have added the following sentence "Our vessel occlusion experiments do not take into account possible anatomical auto-regulating changes that can occur in the system over time. After a certain period, a real vascular system may change or heal to compensate for a blood circulation imbalance. This is a limitation of the model at present. An adaption of the model to allow for vessel tree growth would be highly interesting for instance in infarction modeling, but it is outside the main scope of this paper."

22. *p14 l50: "evacuated" suggests to me that the system is influenced by an active experiment, this is probably not meant here.*

We apologize for the poor term choice. Changed to "drained".

23. *p15 l38: should this read "... frog tongue ..."?*

Corrected

24. *p15 l54: I would recommend not using the term "(in)significant" outside a statistical context to avoid potential confusion.*

Changed to "negligible."

25. *p15 l56: should probably read "... including the venous ... just the arterial ..."*

Revised.

26. *p19: Ref 20 looks incomplete, Ref 29 contains redundant information*

We thank the reviewer for pointing this out. It has been corrected.

27. *Data availability: The images representing the dataset look fine. I did not read the matlab code in all detail; besides the parameter mentioned above, I did not see anything implausible.*

Conflict of interests statement: I suggested to cite work I have authored. Please check and decide for yourself which citations really should be added.

We will gladly read and consider the work mentioned by the reviewer as a reference. However, we are unable to obtain access to the paper's title or author from the provided information. Please send the reference information.

Technical note: the pdf I downloaded for review contained the text twice, once including references, one without. (If I had to guess, the first part is a pdf uploaded by the authors, the second one generated from the also submitted latex source by the submission system?) I only looked at the first part, pages and line numbers above refer to this version.

Reviewer #2

Review — A non-linear multi-scale model for blood circulation in a realistic vascular system

This paper introduces a multi-scale model for blood circulation on a realistic vascular network, illustrated by simulations on a 2D frog tongue. The main novel contribution is the pressure drop at bifurcations and the dependence of vessel radius on pressure (elasticity) providing a good balance between complexity/accuracy and computational cost although inducing nonlinearities. The paper reports promising results for occlusions in root vessels, illustrating the potential of the model in such applications. Nevertheless, I believe efforts need to be made both regarding the description of the model and its validation, as detailed in the following comments.

Materials and methods

1. Figure 1 nicely illustrates the coupling between the components of the model but the (too many) dotted lines are a bit confusing. Even if it is quite obvious, the green dotted lines aren't introduced anywhere.

We thank the reviewer for pointing this out. We have revised and give an additional explanation for the green dotted lines in Figure 1.

Figure 1: Quasi-3D numerical model for a 2D spatial problem. The xy (vertical) axis represents the 2D computational domain in space and the third axis (horizontal) is the model axis. Blood flows from the roots of arterial network through the arterial network (red), then to the continuous domains of the capillary compartments (light red for arterial and light blue for venous compartment), and finally to the venous network structure (blue) and to the venous roots at the bottom of the venous network. The arterial and venous terminal nodes are connected to their respective continuous capillary domains in the capillary bed (red and blue dotted lines). The green dotted lines represent the pixel-wise bridge between capillary compartments, which is modelled as the blood perfusion in Section ...

2. The description of the graph structure model involves quite a lot of notations. I recommend adding a pictorial illustration of the graph to make it clearer.

We have added a new figure (Figure 2) to illustrate the framework for the pressure drop induced by the nonlinearities.

Figure 2: Illustration of pressure drop at junctions, see (??). The arrow represents the direction of blood flow.

3. The results section compares the changes due to each nonlinearity. I believe the description of the model should match this, with the description of the linear model followed by a paragraph for each nonlinearity (from linear to baseline model).

The referee suggests an alternative organization of the result sections. We had a lengthy discussion among ourselves to determine the explanation sequence. Our conclusion was to keep the original organization. We might reconsider if the referee means that this a very crucial point for acceptance of the manuscript.

4. (page 4, line 45) It is not clear what P_{in} and P_{ext} are. Are those given values ? The pressure on the arterial and venous root nodes is also denoted as P_{ext} (2.14), is that the same P_{ext} ? Moreover, (2.14) seems to indicate that the pressure imposed at the arterial roots and venous roots is the same, while Table 1 states two values for artery inlet pressure (30 mmHg) and vein outlet pressure (7.5 mmHg) which looks more coherent.

We appreciate the reviewer bringing this to our attention. P_{in} and P_{ext} in the elasticity equation denotes the blood pressure within the vessel wall and next outside the vessel wall (the capillary). " P_{in} is defined as an average pressure in the segment, that is the average pressures at both segment endpoints, and P_{ext} as the capillary pressure at the midpoint outside the vessel." P_{ext} in (2.14) refers to the boundary condition, which is, indeed, our error. Therefore, we have made changes to use P_{BC} instead, with constant values for artery inlet pressure (30 mmHg) and vein outlet pressure (7.5 mmHg).

5. Regarding Table 1, artery inlet pressure and vein outlet pressure are reported as coming from [22] while the text says pressure outlet in veins comes from [24] (page 9, line 25)

We have corrected the reference in Table 1 to [24].

6. In (2.10), Ω_a and Ω_c are not introduced

We have fixed the error.

7. The paragraph on coupling vascular structure and capillary models is very unclear (whereas crucial for the model):

- a. The finite radius ϵ used with the distribution function is not properly defined. The notation N_ϵ as the number of cells in this radius seems to indicate that it represents a region of radius ϵ around the terminal node? How is it chosen?
- b. The integration domain Ω in (2.12) is not properly defined
- c. Notations aren't consistent : terminal node are denoted i while l is used in (2.13)
- d. The short explanation on tube radius computation is a bit confusing, the notation r_t and r_c aren't used anywhere else in the paper.

We appreciate the reviewer's feedback and have made the necessary changes.

- a. "The finite radius ϵ can be a set to a chosen constant or to a fraction of the terminal vessel's radius. In this manuscript, it is set to three pixels to represent the vessels that are not observable in the imaging data. "
- b. The integration domain Ω in (2.12) is the capillary compartment domain connected to the terminal node, which is the Darcy domain.
- c. We have fixed the error that the terminal nodes are denoted t similar to the matrix explanation in the numerical implementation.
- d. We thank reviewer again for alerting us to this confusing explanation. We decided to change the explanation as follows: "In this work, the resistance was estimated as a constant by using equation (2.1) to compute the resistance of a cylindrical tube connecting a terminal node to the Darcy domain inside a finite radius ϵ . The tube has a length of ϵ and a radius of $8^{-3}r_t$, where 8 is the number of circumference cells within a three-pixel finite radius of ϵ ."

8. (Page 6 line 19) The graph segment is denoted i while it was previously denoted n (on page 4).
Corrected.

9. (Page 6 line 54) Matrix A_D should be A_{D-D} according to notations in the full coupling paragraph (Page 7)

This observation is correct. The misprint has been corrected.

10. (Page 7) Authors state that TPFA method is consistent and convergent on the quadrilateral grid used in this study. I recommend introducing the use of this quadrilateral grid earlier.

We have changed accordingly. The new sentence in the beginning of subsection provide the explanation for quadrilateral grid. "The capillary bed discretizations were obtained from the tissue region of the tongue image shown in Figure 2, and the scanned image resolution was defined as uniform quadrilateral grids in the Darcy domains (see table 1)."

11. (Page 7, full coupling paragraph) The matrices A_{D-T} and A_{T-D} should also be listed, and the b_N term in the RHS is not introduced. The notation F representing the system is confusing : $F(x_n) = b_n$ represents the vascular graph model while $F(x) = b$ represents the full system. Moreover I don't think these F notations are really needed (but if so, these should be properly defined earlier when introducing the model).

This observation is correct. It has been changed and introduced $F(x) = b$ in the vascular graph model implementation as follows: "Finally, the whole is represented as a system of nonlinear equations $\mathbf{F}(\mathbf{x}) = \mathbf{b}$, where $\mathbf{F}(\mathbf{x}) = \mathbf{A}(\mathbf{x})\mathbf{x}$."

12. (Figure 2) The point D mentioned on Page 9 line 18 is not visible on the figure (even if one can guess where it is).

The label in the figure has been changed.

13. (Page 9 line 34) "it was also manually adjusted". I guess it has to do with the permeability. Which value is set then ? And how different this is compared to the estimated permeability ?

We have revised the sentence to: "We obtain an estimate permeability of $1.2 \times 10^{-6} \text{ mm}^2$. However, the value was adjusted in the simulation (refer to table 1) to match a normal blood flow in veins vessels based on ..."

14. There is no information about the software/solver used. The code and data is accessible but there is no instructions on how to run the model and reproduce the results.

We have added a short information, along with an instruction file to run the model and reproduce the results.

Result and discussion

1. *The impact of each of the nonlinearities added to the model (pressure drop at junctions and vessel elasticity) is an important contribution of the paper. Nevertheless, the comparison of the linear, elasticity, junction and baseline models remains a bit short. Table 2 clearly shows that both nonlinearities have an essential role, but the text only describes the content of Table 2. It would be interesting to elaborate further on what (and where) are the differences between the results obtained from junction, elasticity and baseline models.*

We've added running times in Table 2, and some explanation in the text for the running time differences.

2. *The presented results are compared to both modeling and experimental papers, with applications on human cerebral cortex or rats cortex. These comparisons are mentioned all through the result and discussion section, making it hard to follow. Adding a new table summarizing the comparisons that have been made (giving what's in good agreement and what are the differences) would help the reader to have a better understanding of the potential applications of this model and its limitations.*

We have added a table for comparison and differences, with pointers to literature, see Table 3.

3. *The application on occlusion in root vessels gives consistent results and shows again the importance of the nonlinearities. Nevertheless, the radius dilation effect is far from experimental findings (Page 15 line 27). This could be further detailed.*

We appreciate the reviewer's feedback. At present, our model has the limitation that it cannot account for vascular changes over time because of the steady-state assumption. Therefore, we highlight this shortcoming in this following sentences: "The simulation design and parameters are crucial in determining vessel change, which may allow for greater radius dilation similar to the experiment. However, in this framework, a steady-state flow is far from a realistic blood circulation that allows for the simulation of instantaneous changes in the vascular system. The dilation of the upstream vessels is part of the natural self-regulation system that keeps the organ's blood circulation balanced."

Minor comments / typos :

- Page 8 line 58 : *Misplaced comma*
- Page 9 line 6/7 : *of the it limitations*
- Page 14 line 41 : *the blood regulation is highly dependent*
- Page 15 line 29 : *to allows*

Thank you very much. The misprints have been corrected.

Appendix C

Dear editor

Please find enclosed the revised version of the manuscript "A nonlinear multi-scale model for blood circulation in a realistic vascular system". We wish to thank the Referees for a careful reading of the article and for relevant and constructive comments. We have addressed all the comments and concerns to the best of our knowledge. Please refer to the text below for the Editor and Referee comments (in black) in *italic* and our author response (in blue). We hope that the Referee are satisfied with the changes and they find this revised version, suitable for publication in Royal Society Open Science.

Response to the reviewers

We thank the reviewers for a thorough review of the manuscript, and their opinions regarding the content and presentation of the material. The revision of the manuscript is listed according to the reviewer comments.

Reviewer #1

Comments to the Author(s)

The authors have addressed all my comments in their revised submission. I have just a few minor points below, otherwise I recommend to accept the manuscript for publication. In the page and approximate line numbers, I am referring to the version with highlighted changes (second one in the pdf I received from the submission system):

- Title page: says "2014" (probably by the template and to be updated anyway)*
Corrected (it was from the template indeed.)
- p2 l30: "they" is unclear (it is neither the capillaries and arterioles nor the authors of [10]), you'll probably have to repeat "the authors"*
Revised as suggested.
- Fig. 1: Red, blue, and green are indistinguishable in my greyscale printout. I would recommend using a different line style (e.g., dotted) for the green correspondence, as this is technically different from the red and blue ones (for which indistinguishable colors are not a problem).*
We thank the reviewer for pointing this out. We have revised accordingly.
- p3 l54: should read "a homogeneous and isotropic porous medium" (not "an" and singular)*
The sentence is now revised, thanks.
- Fig. 2: I do not think this diagram is helpful, at least not in its present form. If I understand correctly, (2.1) is the formula for a single vessel segment, so three boxes with ΔP_n^h suggests that all three segments at a junction individually have the same pressure drop. Similarly, (2.2) is an additional pressure drop depending on the parent and a single offspring segment, so two boxes with ΔP_n^b on the left hand side suggest that the pressure drop is the same for both offspring segments. On the right hand side, why is there a single ΔP_n^b even though there are two datum vessels? It seems to me that more indices are needed here (dat, n_1, n_2 on the left; dat_1, dat_2, n on the right) and if my understanding of the formula is correct, two $\Delta P_{...}^b$ are needed upstream of the bifurcation? Should (2.2) better be written as $\Delta P_{dat,n}^b$ then?*
Revised as suggested.
- p6 l28: Should this read "user-chosen"?*
This is now corrected.
- Eq. 2.12: could be integrated in the sentence as "according to the relation" (no colon)*
Corrected accordingly, thanks.

8. *p8 l22: should read "non-zero entries" or "non-zeroes" (without "entries"), but I think the latter might be confusing*
We have now changed to "non-zero entries", as suggested.
9. *p8 l55: I would recommend somehow referring to the data availability statement rather than just stating "are available".*
Revised according to the other reviewer suggestion, thanks.
10. *p9 l28f "... optimisation ... linearized system": mix of British and American English (please check entire manuscript, I did not pay attention everywhere)*
We thank the reviewer for pointing this out. We have checked the entire manuscript and revised accordingly.
11. *Fig. 3: Does this reproduced figure require some kind of copyright statement (has probably expired)?*
We believe the picture is in "public domain" due to its age (in Europe 70, years after the artist's death), and is thereby not protected by copyright. We still need to name the author/artist by citing the book.
12. *p10 l25 (and elsewhere in the manuscript): Should the species names be provided in parentheses?*
The species names should be italicized without parentheses according to the International Commission on Zoological Nomenclature.
13. *p10 l37: Does $x \pm y$ mean (arithmetic) mean \pm standard deviation throughout the manuscript? If so, I would recommend stating so at the first occurrence.*
Revised as suggested.
14. *p14 l45 (and in the caption of Table 2): flow resistance is a phenomenon, an electrical resistor is a physical component; I find it a bit confusing to call this an analogy. To me, the corresponding phenomenon is electrical resistance. Similarly, elasticity corresponds to capacitance.*
Revised as suggested.
15. *p15 l35: The difference is about 4.5 orders of magnitude smaller than the value referred to, is this a meaningful difference due to the model, or a rounding effect due to the numerical implementation of the computations?*
The difference is small and the solution is close to the model referred. This is related to the numerical implementation of the computations, as well as the approximation used to solve the non-linear system.
16. *Table 3: Is everything written in this table cited from the respective papers, or are you partially describing the results in your own words? I would suggest clearly marking what was cited here to avoid ambiguity who "We" is. Also, for [46], the right column contains a sentence without verb.*
Revised according to the other reviewer suggestion, thanks.
17. *p18 l56: I would recommend not using "significant" outside statistical analyses.*
Changed to "notable" , thanks.
18. *p20 l25: The manuscript without highlighted changes contains a stray "1pc" before the ethics statement.*
Removed, thanks.
19. *The reference list contains a mix of title case and lower case journal names and a stray "." in [31].*
We have revised accordingly, thanks

Reviewer #2

Many thanks to the authors for their responses to the reviews, and updates to the paper.

1. Thanks for taking into account my suggestion of describing the model with a pictorial illustration (Fig 2), it helps a bit. Nevertheless, the use of the n notation on all segments is not fully accurate. In my understanding, n denotes the different vessel segments. The pressure drops in the figure should then be denoted as $\Delta P_0, \Delta P_1, \dots$. In this regard, the notation r_0 in equation (2.5) is confusing, since it can be interpreted as r_n with $n = 0$ (radius of the first vessel segment), instead of the vessel radius without pressure gradient (which applies to all vessel segments)

Revised as suggested.

2. The updated notations in section 2(e) makes it clearer, especially subscript t for the terminal node. But I am a bit confused with Ω_β , which is defined as “the Darcy capillary volume” (Page 6 line 4, diff file) and as “the capillary compartment domain connected to the terminal node” shortly after (Page 6 line 36). These are the same (right ?), and if so it doesn't need to be re-introduced (in different words).

We thank the reviewer for pointing this out. We have revised accordingly.

3. On Table 2 : I quite like the new version of the table, especially the new computational time column. But are the numbers on that column for the occlusion models t_{tot} or t_{sys} (t_{tot} I guess ?) Could you elaborate on the huge difference in terms of computational time regarding occlusions (baseline model) ? Also I noticed that the results in Table 2 (and earlier in the text) changed. Why is that ? What changed in the model compared to the initial version of the paper ?

We thank the reviewer for pointing this out. The numbers are for t_{tot} in the occlusion models (without parantheses). The large difference is primarily due to a convergence problem with this particular geometry occlusion. Since changes in the permeability value and the method for computing the $kappa$ parameter in equation (2.13), which is linked to model coupling, the results have changed.

4. On Table 3 : This table is very helpful, thanks for adding it. The text in the Object column contains random spaces. Moreover, I would recommend using quotation marks when quoting text from a paper, separating parts of the text that are not contiguous in the original text or using [...]. For example when citing [47] : “We found no evidence for active vasodilation in neighboring arterioles in response to a penetrating arteriole occlusion”, “We observed a slight, but not statistically significant, drop in both RBC speed and in RBC flow in neighboring penetrating arterioles after the occlusion, indicating that blood flow to the area surrounding the occlusion was mildly decreased”, ...

Revised according to the reviewer suggestion, thanks.

5. Thanks for giving details and instructions regarding the provided codes. For what it's worth, I usually add the link to the available code as a reference, so that I can cite this reference in the text (might be useful Page 8, line 56, diff file).

Revised as suggested.

6. Some more minor comments / typos :

- Page 4 line 33 (diff file) : refer j - refers
- Page 6 line 29 : can be a set to j - can be set to
- Page 7, line 7 : For each segment i j - For each segment n
- Page 19, line 8 : had an detectable impact j - had a detectable impact

We thank the reviewer for pointing this out. It has been corrected.